# 🧩 MaintainCoder: Maintainable Code Generation Under Dynamic Requirements

**Zhengren Wang**[†1,3], **Rui Ling**[†1], **Chufan Wang**[†1], **Yongan Yu**[†2]
**Sizhe Wang**[1], **Zhiyu Li**[*3], **Feiyu Xiong**[3], **Wentao Zhang**[*1]
[1]Center for Data Science, Peking University [2]McGill University
[3]Center for LLM, Institute for Advanced Algorithms Research, Shanghai
wzr@stu.pku.edu.cn, {lizy,xiongfy}@iaar.ac.cn, wentao.zhang@pku.edu.cn

## Abstract

Modern code generation has made significant strides in functional correctness and execution efficiency. However, these systems often overlook a critical dimension in real-world software development: *maintainability*. To handle dynamic requirements with minimal rework, we propose **MaintainCoder** as a pioneering solution. It integrates the Waterfall model, design patterns, and multi-agent collaboration to systematically enhance cohesion, reduce coupling, achieving clear responsibility boundaries and better maintainability. We also introduce **Maintain-Bench**, a benchmark comprising requirement changes and novel dynamic metrics on maintenance efforts. Experiments demonstrate that existing code generation methods struggle to meet maintainability standards when requirements evolve. In contrast, MaintainCoder improves dynamic maintainability metrics by more than 60% with even higher correctness of initial codes. Furthermore, while static metrics fail to accurately reflect maintainability and even contradict each other, our proposed dynamic metrics exhibit high consistency. Our work not only provides the foundation for maintainable code generation, but also highlights the need for more realistic and comprehensive code generation research. Resources: https://github.com/IAAR-Shanghai/MaintainCoder.

## 1 Introduction

> *"The Only Constant in Life is Change."* — Heraclitus

The advent of large language models (LLMs) has revolutionized code generation [20, 44, 11], with modern tools like GitHub Copilot and ChatGPT demonstrating remarkable capabilities in synthesizing functionally correct code. Benchmarks like HumanEval [7], MBPP [28], and SWE-Bench [21] have driven these advancements by measuring correctness through static test cases. However, this narrow focus overlooks a critical dimension of real-world software engineering: maintainability—the ability to adapt code to evolving requirements with minimal rework.

**Maintainability Crisis** Heraclitus' philosophy, "The only constant in life is change," reflects a truth painfully evident in software evolution: as user needs evolve, markets shift, and technologies advance, software undergo perpetual changes. This oversight on maintainability leads to the dangerous reality—code that works today but becomes prohibitively expensive to adapt tomorrow [6]. For instance, the collapse of Knight Capital exemplifies this crisis—$440 million lost in minutes due to

---

† Equal contribution; ∗ Corresponding author.
The first author completed this work during an internship at IAAR.

39th Conference on Neural Information Processing Systems (NeurIPS 2025).

unmaintained legacy code [33]; the \$100 billion remediation cost for Year 2000 Problem revealed how short-sighted design decisions create exponential future costs [31]. Industry studies also quantifies the severity: 60–80% of software lifecycle costs stem from post-deployment maintenance [29, 4], due to structural deficiencies like high coupling and low cohesion that amplify rework effort. Another example is online multi-round code generation. Due to ambiguous specifications and human-AI communication barriers, the initial codes usually deviate from final requirements, while subsequent refinements either are prone to errors or inadvertently discard previously satisfied requirements [27].

**Motivation**   Despite taking up the lion's share of software development resources, maintenance is the least studied phase of software development [35, 9]. Although recent benchmarks have expanded evaluations to diverse dimensions like readability and efficiency [43, 18], yet retain static test cases. The static test cases fails to capture the iterative nature of software evolution, and are ill-suited for maintainability evaluation. Moreover, traditional software metrics like cyclomatic complexity and maintainability index, still remain limited to static code analysis, failing to capture the dynamic scenarios. Meanwhile, code generation systems—from foundation models like CodeLLaMA [32] to multi-agent frameworks like SWE-agent [41]—just optimize for task completion but neglect structural qualities critical for maintainability. Therefore, fundamental gaps or challenges persist: (1) No benchmark quantifies maintainability under requirement evolution cycles; (2) No method systematically applies software engineering principles (e.g., design patterns) to enhance maintainability; (3) No discussion on the interaction between correctness and maintainability.

In contrast to naive implementation, structured approaches like the Waterfall model and design patterns are proven to be effective to improve maintainability during decades. For example, Dong et al. [10] confirms the benefits of the waterfall model in initial code generation, with 29.9–47.1% relative improvement on correctness; Hegedűs et al. [13] reveals a very high Pearson correlation of 0.89 between design patterns and software maintainability, via the analysis of more than 300 revisions of JHotDraw system. Given these challenges and opportunities, we propose MaintainBench and MaintainCoder as a paradigm shift toward maintainable code generation, bridging the gap between static correctness and dynamic maintainability.

**Contributions**   Our main contributions are threefold:

- **Dynamic Benchmark**: We introduce MaintainBench, the first benchmark assessing code maintainability through requirement evolution cycles. Constructed through systematic extension of well-established benchmarks (HumanEval, MBPP, APPS, CodeContest, xCodeEval), it incorporates diverse requirement changes with expert-curated test cases. Novel metrics quantify modification efforts and structural adaptability dynamically.

- **Systematic Generation**: We introduce MaintainCoder as a pioneering solution. It integrates the waterfall model with multi-agent collaboration and classical design patterns. Specialized agents conduct requirements analysis, modular decomposition, and pattern application to enforce high cohesion, low coupling, single responsibility, and maintainability.

- **Empirical Insights**: Extensive experiments reveal that (1) Previous methods suffer from significant maintainability degradation under dynamic requirements (low pass rates and high code changes), even for multi-agent systems such as AgentCoder and MapCoder. (2) MaintainCoder not only improves maintainability metrics by up to +60%, but also enhances the correctness of initial codes robustly. (3) Static metrics not only fail to accurately reflect maintenance efforts, but also exhibit contradictory trends among different metrics. In contrast, the dynamic metrics proposed by MaintainBench are more effective and consistent.

## 2   Related Work

**Code Generation Benchmarks**   Existing research on code generation has been propelled by benchmarks emphasizing functional correctness. Pioneering works span multiple levels of complexity: function-level tasks, such as HumanEval [8] and MBPP [3]; competition-level problems, as addressed by APPS [14] and CodeContests [23]; and repository-scale challenges tackled by SWE-Bench [21] and RepoBench [25]. Despite these advancements, these benchmarks predominantly use static test cases measuring correctness through execution pass rates. Although recent efforts, such as RACE [43] and EffiBench [18], have expanded the evaluation to include readability and efficiency, yet remain

constrained by static evaluation, failing to capture the dynamic nature of software development. Even SWE-Bench, which evaluates dynamic problem-solving via GitHub issue resolution, focuses on challenging task completion capabilities rather than assessing code maintainability itself—i.e., generating inherently maintainable code from the outset. MaintainBench bridges this critical gap with metricization of code maintainability through requirement evolution cycles, where 80% of software costs occur in real-world applications [29, 4, 5].

**Code Generation Systems**   Modern code generation systems have evolved from three key aspects: foundation models, instruction-tuned variants, and multi-agent frameworks. Foundation models such as AlphaCode [23] and CodeLLaMA [32], established fundamental code synthesis capabilities through large-scale pretraining on code corpora. Building on these foundations, instruction-tuned variants like WizardCoder [26] and Magicoder [38] improved context understanding through data distillation and complex instruction augmentation. The frontier now shifts to multi-agent frameworks: MetaGPT [15] and ChatDev [30] employ standardized outputs to coordinate multiple agents, while SWE-agent [41] and OpenDevin [37] focus on repository-level task decomposition. AgentCoder [17] and MapCoder [19] introduce specialized testing agents and retrieval process respectively. EvoMAC [16] proposes self-evolving multi-agent collaboration networks to optimize code through multiple iterations. However, all of them overlook maintainability, and are prone to generate brittle code that is difficult to maintain. MaintainCoder bridges this gap by integrating principles like the waterfall model and classical design patterns.

# 3   Method

## 3.1   Problem Formulation

**Task Definition**   Previous code generators optimize for immediate correctness but neglect software engineering's long-term challenge: evolving problem series $\{P_0, P_1, ..., P_n\}$ drive iterative code adaptations $\{C_0, C_1, ..., C_n\}$, which incurs maintenance cost $\mathcal{M}(C_i \rightarrow C_{i+1})$. Current approaches treat each $P_i$ independently, and ignore structural deficiencies that accrue *technical debt*. We investigate code generator $\mathcal{G}$ producing $C_0 = \mathcal{G}(P_0)$ with both high correctness and maintainability.

**Maintainability Measurement**   Previous works measure maintainability through static analysis indexes, such as Halstead volume [12], cyclomatic complexity or maintainability index. However, the ultimate goal of maintainability is to reduce maintenance efforts. We thus formalize dynamic maintainability through cumulative maintenance efforts with discount factor $\gamma$.

$$\mathcal{M}(C_0) = \mathbb{E}\left[\sum_{i=0}^{n-1} \gamma^i \mathcal{M}(C_i \rightarrow C_{i+1})\right]$$

Given maintainability's intrinsic nature, we posit that limited probing can substitute unlimited requirement evolution simulations. We employ Monte Carlo estimation and first-order truncation as the proxy to address infinite expectation horizons and persistent requirement changes, which reduces both computational cost and estimation variance, yielding $\hat{\mathcal{M}}(C_0) \approx \frac{1}{N}\sum_{j=1}^{N}\mathcal{M}(C_0 \rightarrow C_1)$.

## 3.2   MaintainBench: A Dynamic Benchmark for Code Maintainability

In this section, we introduce MaintainBench, a novel benchmark designed to evaluate code maintainability in response to evolving software requirements. Unlike traditional code generation benchmarks that focus solely on correctness, MaintainBench evaluates how effectively models can adapt existing code to meet changing requirements. MaintainBench consists of five carefully curated datasets: HumanEval-Dyn, MBPP-Dyn, APPS-Dyn, CodeContests-Dyn, and xCodeEval-Dyn, comprising over 500 Python programming data of diverse difficulty levels. Each extends established benchmarks with systematic requirement changes. The overview of construction process is shown in Fig. 1.

### 3.2.1   Data Selection and Preprocessing

MaintainBench comprises code samples of varying difficulty levels, roughly entry-level, mixture-level, and competition-level. Specifically, it extends five widely-used code generation benchmarks: HumanEval [7], MBPP [28], APPS [14], xCodeEval [22], and CodeContests [23].

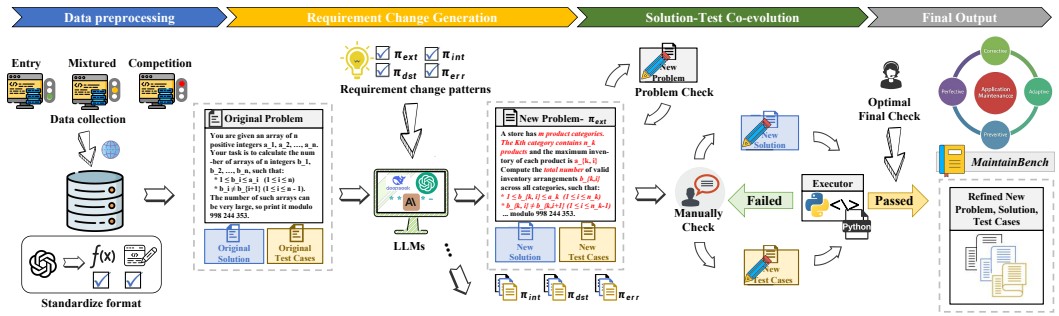

Figure 1: The systematic construction of MaintainBench. 1) Data preprocessing selects and curates problems from existing benchmarks into entry-level, mixture-level, and competition-level difficulties. 2) Requirement change generation applies four systematic patterns (functional extensions, interface modifications, data structure transformations, and error handling enhancements) to create evolved problem variants. 3) Solution-test co-evolution refines solutions and test cases together through iterative execution and refinement. 4) Quality check employs both automated test and multi-stage expert review to ensure reliability, correctness, and alignment with intended requirement changes.

**Entry Level**   For entry-level difficulty, we select problems from the HumanEval and MBPP datasets, which are designed to be solvable by newbie programmers. These datasets are widely used to evaluate the code generation capabilities of LLMs, covering topics such as language comprehension, algorithms, basic mathematics, programming fundamentals, and standard library functionality. Each problem includes a task description, a reference solution, and a set of automated test cases. We sample 30 problems from each dataset randomly, and extend them to 120 new problems by systematically modifying their requirements.

**Mixture Level**   For mixture-level difficulty, we extend the APPS dataset, which contains problems from introductory to collegiate competition levels. For example, problems from Kattis[1] with difficulty <3 are introductory, 3–5 are interview-level, and >5 are competition-level. We start with a random subset of 50 problems and expand them to over 200 new problems. We standardize the dataset by converting original solutions to functional form and test cases to Python assertions. E.g., to avoid the keyboard input, we utilize GPT-4o to generate functional implementations with manual review.

**Competition Level**   The competition-level dataset includes problems from CodeContests and xCodeEval. CodeContests focuses on competitive programming tasks, while xCodeEval provides a large-scale executable dataset with varying levels of difficulty. To distinguish this level from entry-level and mixture-level tasks, we select 30 high-difficulty problems from each dataset and extend them to over 120 new problems. As with the mixture-level dataset, we standardize the data format using LLM-generated solutions, followed by manual verification to ensure correctness.

### 3.2.2   Extended Data Generation

Building on our preprocessed datasets, the core innovation of MaintainBench is the systematic application of four requirement change patterns, formalized as:

$$P_1, S_1', T_1' = \text{Modify}(P_0, S_0, T_0, \pi_i), \quad \forall \pi_i \in \Pi_B$$
$$P_1, S_1, T_1 = \text{Refine}(P_1, S_1', T_1')$$

where $P_0$, $S_0$, and $T_0$, represent the original problem, reference solution, and test cases, respectively. The $\text{Modify}(P_0, S_0, T_0, \pi_i)$ applies one of four requirements change patterns $\pi_i$ from our pattern set:

$$\Pi_B = \{\pi_{\text{ext}}, \pi_{\text{int}}, \pi_{\text{dst}}, \pi_{\text{err}}\}$$

, and generates an initial extended versions. This transformation process is designed to simulate realistic software evolution scenarios where developers must modify existing code to accommodate

---

https://open.kattis.com/

changing requirements rather than implementing solutions from scratch. The refinement process co-evolves both new solution $S_1$ and test cases $T_1$ with automatic evaluation and manual curation loops, ultimately yielding the final reference solution and test cases for the modified requirement $\pi_i$.

**Requirement Change Generation**     According to the ISO/IEC/IEEE 14764:2022 specification [34], software maintenance can be roughly classified into four types: corrective maintenance, preventive maintenance, adaptive maintenance, perfective maintenance, where the latter two categories comprise about 80 percent of software maintenance [36]. Hence, we craft four distinct requirement change patterns to capture different software evolution scenarios:

- *Functional Extensions*[$\pi_{\text{ext}}$]: Functional extensions increase complexity by introducing realistic additional requirements while preserving the relevance with the original problem. Effective solutions must both comprehend the original code and appropriately extend its problem-solving capabilities. For example, the extended problem should involve self-invoking [42], i.e. calls to the existing functions, modeling the interactions between functions or classes. Functional extensions cover both adaptive and perfective maintenance.

- *Interface Modifications*[$\pi_{\text{int}}$]: Interface modifications further enhance the diversity of generated problems, and examine the adaptability of original solutions in protocol dimension. E.g., the API changes caused by the update of the external dependency library. The modified problem remains closely related to the original, but introduces changes to input parameters, return types, or other interface components to assess the robustness and flexibility of solutions. Interface modifications correspond to adaptive maintenance.

- *Data Structure Transformations*[$\pi_{\text{dst}}$]: Data structure transformations require the generated problem to adopt different data structures where the description explicitly specifies the modifications of the data representation. This transformation mimics real-world scenarios in which software systems evolve to accommodate efficiency, scalability, or compatibility constraints. Data structure transformations primarily align with perfective maintenance.

- *Error Handling Enhancements*[$\pi_{\text{err}}$]: Error handling enhancements explicitly introduce required error-handling mechanisms, which capture specific error types such as `ZeroDivisionError`, `IndexError`, and other problem-specific exceptions. The enhanced new problem better reflects the fundamental aspect of reliability and maintainability. Error handling enhancements encompass corrective, preventive, and perfective maintenance.

**Solution-Test Co-evolution**     We utilize GPT-4o to generate initial versions of each transformation, including preliminary solution $S_1'$ and test cases $T_1'$, followed by an iterative refinement process as validation. That is, we execute $S_1'$ against $T_1'$ using a `Python` interpreter to assess correctness. For any failing tests, we conduct manual expert reviews to diagnose and resolve issues in both the solution and test cases, refining $S_1'$ into $S_1$ and $T_1'$ into $T_1$. If discrepancies persist, the execution-review cycle is repeated until all test cases pass successfully.

More details on solution-test co-evolution and quality check are left in Appendix B for space reason.

### 3.3   MaintainCoder: A Multi-Agent System for Maintainable Code Generation

As shown in Fig. 2, MaintainCoder delivers highly maintainable codes via a novel multi-agent system that mirrors the human software development lifecycle. It designs an orchestrated pipeline of LLM agents, each specializing in distinct phases of software development while maintaining contextual awareness through inter-agent communication empowered by AutoGen framework [39].

#### 3.3.1   Code Framework Module

This module transforms user requirements into maintainable architectural blueprints.

**Requirements Analysis Agent**     The requirements analysis agent is tasked with in-depth analysis of software requirements. It decomposes the problem step by step through chain-of-thoughts, prioritizing user goals and rethinking practical constraints. The agent receives user requirements, extracts pivotal goals, identifies core functions, highlights key challenges, and proposes high-level solutions. The output analysis report provides concise guidance for follow-up agents. This analysis process avoids unnecessary complexity and focuses on high-priority tasks and high-level computational logic.

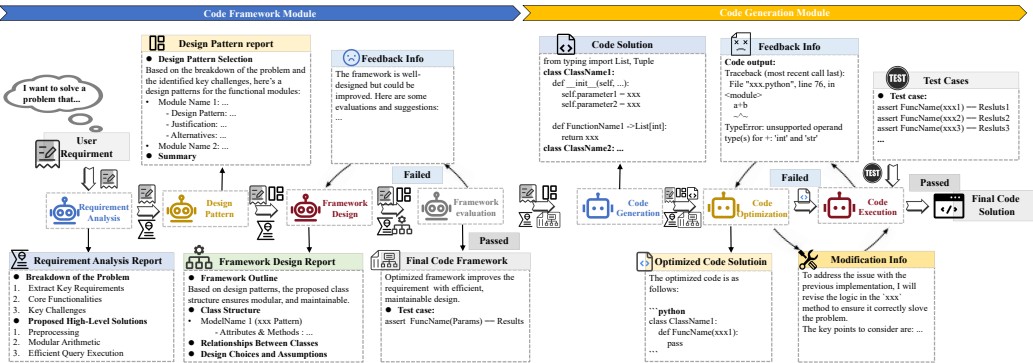

Figure 2: Overview of MaintainCoder. MaintainCoder features best practices like the Waterfall model, classical design patterns, and iterative refinement mechanisms for critical stages. 1) The Code Framework Module systematically transforms user requirements into an optimized, maintainable architectural blueprint; 2) The Code Generation Module implements the blueprint into production-ready code, rigorously adhering to both the framework specifications and initial user requirements.

**Design Pattern Selection Agent**    The design pattern selection agent serves as the software architect, focusing on selecting appropriate design patterns for each functional module. It receives functional modules and key challenges from the requirements analysis agent, evaluates the overall architecture and module interactions, and selects the most suitable design pattern. This selection prioritizes patterns that promote modularity, reduce coupling, and enhance scalability and reusability. The agent also suggests alternative patterns with corresponding applicable scenarios. The output is a structured list of modules, each including the module name, main design pattern, reasons for selection, and alternative patterns with applicable scenarios.

**Framework Design Agent**    The framework design agent agent constructs a modular and robust code framework based on functional modules and selected design patterns. This agent not only designs a preliminary class structure adhering to single-responsibility principles, but also clarifies dependencies to enhances modularity, reusability, and maintainability. It will revise the framework in response to feedback from the framework evaluation agent.

**Framework Evaluation Agent**    Given user requirements, analysis reports, and class structures/relationships as inputs, it reviews design clarity, scalability, performance, and conformance to best practices. This agent identifies issues related to coupling, cohesion, and reusability, avoids overly fine-grained modules, i.e. over-fragmentation. Finally, it delivers actionable suggestions focused on major improvements that impact performance and scalability, which will be returned to the framework design agent for improvement loop.

### 3.3.2    Code Generation Module

This module implements blueprints into executable codes, also guided by requirement analysis.

**Code Generation Agent**    The code generation agent converts detailed framework designs into complete, functional Python code. It ensures readability and maintainability with clear comments and documentation. By carefully analyzing the original requirements, requirements analysis and final framework design, it creates code structures conforming to PEP 8 and PEP 257 specifications. Comments will focus on code purpose, design choice, and especially non-obvious logic, rather than redundantly describing the obvious content. The generated code then includes appropriate class structures, methods, and uniformly named interface functions for testing. After the initial generation, a test sample selected from the test set is inserted as assertion for iterative debugging. The overall goal is to produce fully functional, understandable, and expandable code.

**Code Optimization Agent**    The code optimization agent refines the generated code to meet expected functional and performance requirements. By analyzing the problem requirements, framework design,

original code and test cases, this agent can identify potential problems or room for improvement. First, the agent is forced to thoroughly understand the user requirements and framework design to ensure modifications align with the overall intent. Next, it reviews the original code for logic correctness and interface functions, creating any missing ones. Then, the code is executed to gather feedback on syntax errors, test failures, unexpected behaviors, etc. Through analysis in chain-of-thoughts, the agent diagnoses the root causes, makes necessary modifications (fixing syntax errors, adjusting logic, handling boundaries, optimizing performance), and retests. This iterative process continues until the code meets requirements and passes tests, ensuring functional, efficient, and reliable solutions.

# 4 Experiments

**Experimental Setup**    For a holistic assessment of maintainable code generation capabilities, we primarily utilized the MaintainBench benchmark introduced in this work, as well as the original datasets with various metrics. Experiments were conducted in two phases. In phase I, we generate initial code $C_0$ for the original problem $P_0$ using MaintainCoder or baseline methods. During this phase, static maintainability metrics were recorded; In phase II, we keep the fixed generator, e.g. GPT-4o-mini, to dynamically probe the maintainability of $C_0$. Specifically, we generate modified codes reflecting different types of changes: $C_0 \rightarrow \{C_{ext}, C_{int}, C_{dst}, C_{err}\}$. These variants were then evaluated to compute the dynamic maintainability metrics. For more reliability and statistical significance, we report Pass@k up to k=5. Please refer to the Appendix D for more details.

**Baselines & Metrics**    For a comprehensive evaluation, we conduct experiments across diverse baselines including GPT-4o-mini, DeepSeek-V3, Claude-3.5-Sonnet, Claude-3.7-Sonnet, Gemini-2.5-Flash-Preview, GPT-4o, and Qwen-Plus [24, 2, 1, 40], as well as multi-agent frameworks such as AgentCoder [17] and MapCoder [19]. We consider GPT-4o-mini and DeepSeek-V3 as the backbones, to highlight the benefit of MaintainCoder and explore the performance boundaries respectively. We also investigate Chain-of-Thought (CoT) and Self-Planning (Plan), which incentivize reasoning or planning capabilities through explicit prompting engineering. We measure static maintainability with the Maintainability Index (MI) and Cyclomatic Complexity (CC). We proposed several dynamic metrics, including *post-modification functional correctness* (Pass@k), the likelihood of generating a correct solution after modifications; *code change volume* ($Code_{diff}$), the percentage or absolute value of modified lines; and *syntax tree similarity* ($AST_{sim}$), the structural similarity between the original and modified abstract syntax trees.

## 4.1 Main Experiments

Generally, MaintainCoder exhibits superior performance across different difficulty levels, with its advantage becoming more pronounced as the complexity increases. Due to page limit, the results on entry-level datasets and case studies are left in the Appendix E for interested readers.

**Performance on Mixture-Level Dataset**    Table 1 reports both static and dynamics metrics on the mixture-level APPS-Dyn benchmark. For static structure, MaintainCoder achieves higher MI scores than all baselines, while cyclomatic complexities are around 3—3x lower than competing models. For dynamic metrics, it also shows significant advantages: MaintainCoder(GPT-4o-mini) attains 50.5% Pass@5 accuracy (15-30 points higher than other variants), 0.797 AST similarity (outperforming the second-best by +28%), and minimizes the relative code differences (29.4%). These results highlight MaintainCoder's unique strength, surpassing advanced models like GPT-4o and Claude-3.7-Sonnet.

**Performance on Competition-Level Dataset**    As shown in Tab. 2, the conclusions on mixture-level or competition-level datasets are largely consistent. For example, MaintainCoder(DeepSeek-V3) attains 36.7% Pass@5 accuracy (outperforming the second-best by +60%), 0.785 AST similarity, and the lowest relative code changes 33.0%. Notably, MaintainCoder retains low code complexity even on challenging problems, i.e. it keep CC scores around 3 while baselines inflate code complexity significantly. Surprisingly, multi-agent systems like AgentCoder and MapCoder, despite optimizing single-round correctness, could impair long-term maintainability and degrade second-turn Pass@k. Moreover, the impact of prompt engineering methods, such as CoT and Plan, appears to be inconsistent or even random, which highlights the robustness and superiority of our MaintainCoder. Finally, we advocate that absolute code changes are less indicative of maintenance costs than relative

Table 1: Maintainability evaluation on mixture-level APPS-Dyn problem set. The best performance is indicated in **bold**, while the second-best performance is in underline.

| Model | Static metrics | | Dynamic metrics | | | |
|---|---|---|---|---|---|---|
| | MI↑ | CC↓ | Pass@5 (%)↑ | $\text{AST}_{sim}$↑ | $\text{Code}_{diff}^{per}$ (%)↓ | $\text{Code}_{diff}^{abs}$↓ |
| **APPS-Dyn** | | | | | | |
| GPT-4o-mini | 63.3 | 5.10 | 35.5 | 0.589 | 140 | **17.0** |
| GPT-4o-mini$_{\text{CoT}}$ | 63.2 | 5.10 | 32.5 | 0.593 | 140 | 17.1 |
| GPT-4o-mini$_{\text{Plan}}$ | 59.2 | 5.01 | 34.0 | 0.618 | 93.3 | 17.3 |
| AgentCoder (GPT-4o-mini) | 63.3 | 5.81 | 21.0 | 0.510 | 66.3 | 20.1 |
| MapCoder (GPT-4o-mini) | 67.8 | 5.98 | 30.5 | 0.583 | 73.8 | 19.2 |
| MaintainCoder (GPT-4o-mini) | **69.5** | **2.75** | **50.5** | **0.797** | **29.4** | 19.0 |
| DeepSeek-V3 | 61.8 | 7.59 | 59.5 | 0.598 | 131 | 22.6 |
| DeepSeek-V3$_{\text{CoT}}$ | 61.3 | 7.28 | 52.0 | 0.634 | 119 | **17.9** |
| DeepSeek-V3$_{\text{Plan}}$ | 59.2 | 6.06 | 54.5 | 0.577 | 130 | 20.6 |
| AgentCoder (DeepSeek-V3) | 60.5 | 6.68 | 48.5 | 0.601 | 104 | 20.5 |
| MapCoder (DeepSeek-V3) | 59.3 | 8.76 | 48.0 | 0.659 | 83.0 | 19.1 |
| MaintainCoder (DeepSeek-V3) | **62.4** | **3.21** | **62.5** | **0.828** | **29.2** | 18.6 |
| GPT-4o | **63.0** | **4.58** | 39.5 | 0.556 | 140 | 17.6 |
| Qwen-Plus | 61.2 | 5.61 | 43.5 | 0.638 | 137 | 17.3 |
| Gemini-2.5-Flash-Preview | 59.7 | 9.00 | **51.0** | 0.631 | 108 | **17.0** |
| Claude-3.5-Sonnet | 60.8 | 4.63 | 47.0 | **0.650** | 103 | 17.3 |
| Claude-3.7-Sonnet | 59.3 | 6.65 | 48.5 | 0.620 | **85.2** | 18.6 |

Table 2: Maintainability evaluation on competition-level CodeContests-Dyn and xCodeEval-Dyn problem sets. The best performance is in **bold**, while the second-best one is in underline.

| Model | Static metrics | | Dynamic metrics | | | |
|---|---|---|---|---|---|---|
| | MI↑ | CC↓ | Pass@5 (%)↑ | $\text{AST}_{sim}$↑ | $\text{Code}_{diff}^{per}$ (%)↓ | $\text{Code}_{diff}^{abs}$↓ |
| **CodeContests-Dyn** | | | | | | |
| GPT-4o-mini | 57.8 | 6.06 | 24.2 | 0.661 | 90.1 | 16.6 |
| GPT-4o-mini$_{\text{CoT}}$ | 58.4 | 5.85 | 23.5 | 0.639 | 96.2 | 17.0 |
| GPT-4o-mini$_{\text{Plan}}$ | 54.6 | 6.62 | 28.8 | 0.716 | 65.6 | 16.7 |
| AgentCoder (GPT-4o-mini) | 62.5 | 7.28 | 18.2 | 0.629 | 44.9 | 18.5 |
| MapCoder (GPT-4o-mini) | **66.1** | 7.32 | 25.0 | 0.689 | 47.5 | 19.0 |
| MaintainCoder (GPT-4o-mini) | 65.8 | **2.68** | **32.6** | **0.833** | **23.2** | 17.4 |
| DeepSeek-V3 | 55.7 | 6.80 | 26.5 | 0.718 | 87.2 | **17.5** |
| DeepSeek-V3$_{\text{CoT}}$ | 53.1 | 11.7 | 20.5 | 0.690 | 52.3 | 21.7 |
| DeepSeek-V3$_{\text{Plan}}$ | 51.4 | 12.2 | 17.4 | 0.614 | 67.6 | 26.3 |
| AgentCoder (DeepSeek-V3) | 43.8 | 19.6 | 16.7 | 0.670 | 48.4 | 28.0 |
| MapCoder (DeepSeek-V3) | 49.0 | 14.6 | 11.4 | 0.655 | 53.0 | 24.8 |
| MaintainCoder (DeepSeek-V3) | **63.1** | **3.64** | **37.9** | **0.788** | 43.2 | 18.5 |
| GPT-4o | **57.2** | 5.28 | 25.0 | 0.622 | 94.2 | 18.1 |
| Qwen-Plus | 52.1 | 6.85 | **28.8** | 0.738 | 72.9 | 18.3 |
| Gemini-2.5-Flash-Preview | 51.4 | 8.48 | 18.9 | 0.714 | 61.0 | 19.3 |
| Claude-3.5-Sonnet | 55.3 | 6.74 | 23.5 | **0.739** | **51.3** | **17.5** |
| Claude-3.7-Sonnet | 53.9 | 8.74 | 24.2 | 0.620 | 85.2 | 18.6 |
| **xCodeEval-Dyn** | | | | | | |
| GPT-4o-mini | 56.1 | 6.44 | 23.4 | 0.651 | 81.5 | 18.6 |
| GPT-4o-mini$_{\text{CoT}}$ | 55.9 | 6.44 | 18.8 | 0.644 | 82.4 | 18.4 |
| GPT-4o-mini$_{\text{Plan}}$ | 51.1 | 6.25 | 22.7 | 0.705 | 53.0 | **18.0** |
| AgentCoder (GPT-4o-mini) | 60.9 | 8.34 | 18.0 | 0.624 | 42.3 | 22.3 |
| MapCoder (GPT-4o-mini) | **63.6** | 8.15 | 15.6 | 0.702 | 43.2 | 21.2 |
| MaintainCoder (GPT-4o-mini) | 60.5 | **2.72** | **27.3** | **0.837** | **21.6** | 19.8 |
| DeepSeek-V3 | 54.4 | 7.03 | 21.9 | 0.636 | 92.2 | **21.2** |
| DeepSeek-V3$_{\text{CoT}}$ | 56.7 | 11.6 | 22.7 | 0.613 | 52.0 | 24.6 |
| DeepSeek-V3$_{\text{Plan}}$ | 56.9 | 11.8 | 21.1 | 0.626 | 69.3 | 27.6 |
| AgentCoder (DeepSeek-V3) | 42.2 | 15.2 | 11.7 | 0.690 | 60.0 | 32.8 |
| MapCoder (DeepSeek-V3) | 47.9 | 15.1 | 11.7 | 0.655 | 52.8 | 29.0 |
| MaintainCoder (DeepSeek-V3) | **57.9** | **3.47** | **36.7** | **0.785** | **33.0** | 21.8 |
| GPT-4o | 53.7 | **4.33** | **32.8** | 0.680 | 67.2 | **17.8** |
| Qwen-Plus | 50.7 | 7.24 | 23.4 | 0.684 | 70.3 | 20.4 |
| Gemini-2.5-Flash-Preview | 48.6 | 7.64 | 22.7 | 0.750 | 49.2 | 20.7 |
| Claude-3.5-Sonnet | 50.4 | 6.43 | 17.2 | **0.779** | **41.7** | 18.4 |
| Claude-3.7-Sonnet | **54.0** | 8.64 | 21.9 | 0.734 | 50.1 | 21.4 |

code changes. For example, the software refactoring often inflate code lines but improves long-term maintainability. Previous methods compress lines but led to tangled and unmaintainable code.

## 4.2 Analysis and Discussion

**Correctness versus Maintainability**   Ideally, effective code design paradigms can enhance not only maintainability but also the correctness of initial code. We evaluate MaintainCoder's Pass@5 on the original problems. The results presented in Table 3 validate our hypothesis: MaintainCoder demonstrates improvements in functional correctness, with this benefit becoming more pronounced as problem complexity increases.

Table 3: Pass@5 on the original datasets. Maintain-Coder improves the correctness of initial codes.

| Model | APPS | CodeContests | xCodeEval |
|---|---|---|---|
| GPT-4o-mini | 44% | 18% | 46% |
| MaintainCoder (GPT-4o-mini) | **48%** | **23%** | **57%** |
| DeepSeek-V3 | 66% | 48% | 75% |
| MaintainCoder (DeepSeek-V3) | **69%** | **51%** | **77%** |
| GPT-4o | 48% | 30% | 68% |
| Qwen-Plus | 56% | 30% | 63% |
| Gemini-2.5-Flash-Preview | 65% | 53% | 74% |
| Claude-3.7-Sonnet | 55% | 35% | 58% |

For example, MaintainCoder (GPT-4o-mini) achieves an 11% increase on xCodeEval (from 46% to 57%), which is over double the improvement seen on APPS (44% to 48%). Notably, these gains occur even on already high-performing models like DeepSeek-V3.

**Static Metrics versus Dynamic Metrics**   For static metrics, CC score measures program control flow, Halstead Volume measures arithmetic logic operations, and MI score integrates them. However, these metrics fail to reflect high-level maintainability. For example, AgentCoder and MapCoder achieve high MI metrics, while the second-round correctness was reduced. More confusingly, the changes in MI and CC metrics are contradictory. For instance, the MI and CC metrics of MapCoder(GPT-4o-mini) are 66.1 and 7.32 respectively, both increasing GPT-4o-mini's 57.8 and 6.06. This inconsistency is also evident on AgentCoder. In contrast, dynamic metrics such as Pass@k, $\text{Code}_{diff}$, and $\text{AST}_{sim}$ demonstrate significant consistency. This mu-

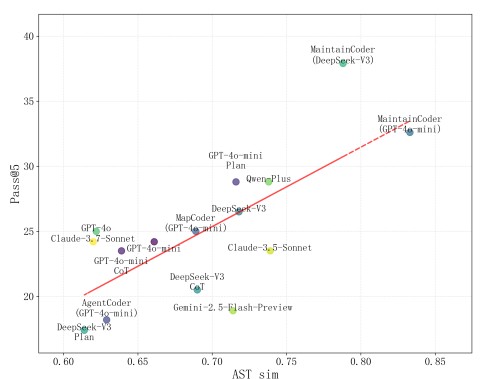

Figure 3: Scatter plot of Pass@5 and $\text{AST}_{sim}$.

tual corroboration advocates that dynamic metrics reflect the maintainability more accurately. Thanks to the construction of MaintainBench, the calculation of dynamic metrics has become possible.

**Ablation Studies**   We conduct ablation studies to investigate the impact of framework evaluation and code optimization. As shown in Table 4, both modules significantly enhance the second-turn correctness, with the code optimization contributing more substantially. For instance, omitting framework evaluation reduces Pass@5 by 25.71% relative points while code optimization incurs more pronounced 37.13% performance degradation. Notably, framework evaluation typically involves only one iteration, which serves as a lightweight yet impactful component.

Table 4: Ablations on Framework Evaluation and Code Optimization. Both agentic modules contribute to final performance. "Perf. Drop" denotes relative performance degradation of Pass@k.

| Base Model | Framework Evaluation | Code Optimization | APPS-Dyn | | CodeContests-Dyn | | xCodeEval-Dyn | |
|---|---|---|---|---|---|---|---|---|
| | | | Pass@5 (%) | Perf. Drop (%) | Pass@5 (%) | Perf. Drop (%) | Pass@5 (%) | Perf. Drop (%) |
| **GPT-4o-mini** | ✗ | ✓ | 49.00 | 2.97 | 31.82 | 2.33 | 20.31 | 25.71 |
| | ✓ | ✗ | 40.00 | 20.79 | 28.03 | 13.97 | 17.19 | 37.13 |
| | ✓ | ✓ | 50.50 | - | 32.58 | - | 27.34 | - |
| **DeepSeek-V3** | ✗ | ✓ | 61.00 | 2.40 | 33.33 | 11.99 | 33.59 | 8.52 |
| | ✓ | ✗ | 48.50 | 22.40 | 25.76 | 31.98 | 28.13 | 23.39 |
| | ✓ | ✓ | 62.50 | - | 37.87 | - | 36.72 | - |

**Human Baseline, Reasoning Models and Multi-Agent Systems**   We benchmarked MaintainCoder against skilled human programmers, the o3-mini reasoning model, and other multi-agent systems such as MetaGPT and EvoMAC. For human baseline, we have recruited participants with competitive

programming experience (Codeforces ratings 1700-2300) to complete tasks from the CodeContests-Dyn benchmark in 30 minutes per problem. As shown in Table 5, the code produced by human programmers demonstrated poorer maintainability metrics compared to AI-generated code. Even skilled developers, particularly under the pressure of problem-solving common in real-world scenarios, can produce less maintainable code. This underscores the value of MaintainCoder in systematically enhancing this vital software quality attribute. Furthermore, MaintainCoder provides significant maintainability enhancements even when applied to a strong reasoning model. It consistently outperforms other multi-agent systems. While frameworks like AgentCoder and EvoMAC focus on solving the immediate problem, MaintainCoder's proactive architectural design, which explicitly embeds software engineering principles, leads to superior long-term maintainability.

Table 5: Comparison with human programmers, reasoning models and more multi-agent systems on CodeContest-Dyn. MaintainCoder demonstrates superior maintainability. MetaGPT requires a stronger base model like GPT-4.1-mini for its tool-calling capabilities

| Model | Static metrics | | Dynamic metrics | | | |
|---|---|---|---|---|---|---|
| | MI↑ | CC↓ | Pass@5 (%) ↑ | $AST_{sim}$ ↑ | $Code_{diff}^{per}$ (%)↓ | $Code_{diff}^{abs}$ ↓ |
| Human Programmers | 53.2 | 8.17 | 23.5 | 0.541 | 112.3 | 23.3 |
| GPT-4o-mini | 57.8 | 6.06 | 24.2 | 0.661 | 90.1 | **16.6** |
| AgentCoder (GPT-4o-mini) | 62.5 | 7.28 | 18.2 | 0.629 | 44.9 | 18.5 |
| MapCoder (GPT-4o-mini) | **66.1** | 7.32 | 25.0 | 0.689 | 47.5 | 19.0 |
| EvoMAC (GPT-4o-mini) | 62.6 | 5.18 | 26.5 | 0.685 | 60.1 | 20.0 |
| MetaGPT (GPT-4.1-mini) | 55.2 | 7.63 | 30.3 | 0.760 | 44.3 | 18.6 |
| MaintainCoder (GPT-4o-mini) | 65.8 | **2.68** | **32.6** | **0.833** | **23.2** | 17.4 |
| o3-mini | 52.1 | 11.3 | 30.3 | 0.661 | 101.6 | 32.9 |
| MaintainCoder (o3-mini) | **62.3** | **3.85** | **36.4** | **0.794** | **27.8** | **21.3** |

**Computational Cost Analysis** We acknowledge that a multi-agent pipeline incurs a higher initial computational cost. However, we argue this is a justified trade-off for substantially improved long-term maintainability and higher initial correctness. As shown in Table 6, MaintainCoder's token consumption is comparable to other multi-agent frameworks like MapCoder and even the reasoning model o3-mini, while being significantly lower than systems like MetaGPT and ChatDev. This initial investment leads to a drastic reduction in subsequent human effort required for debugging and refactoring. We provide a breakdown of token usage in Appendix E for interested readers.

Table 6: Total token usage. The cost of MaintainCoder is comparable to other multi-agent systems.

| Dataset | MaintainCoder | MapCoder | o3-mini | MetaGPT/ChatDev | GPT-4o-mini |
|---|---|---|---|---|---|
| CodeContests | 33.1k | 38.7k | 20.8k | 50k+ | 2.5k |
| xCodeEval | 29.6k | 23.5k | 21.2k | 50k+ | 2.3k |

**Robustness w.r.t. Pass@k & Phase II generator** For a more reliable evaluation, we examine MaintainCoder's robustness with respect to varying k values of Pass@k and different Phase II generators that conduct modifications on initial codes. We leave the experiments in the Appendix E.

## 5 Conclusion

In this paper, we introduced MaintainCoder and MaintainBench for maintainable code generation. MaintainBench, the first benchmark dedicated to dynamic maintainability evaluation, covers diverse and systematic modification scenarios and provides a standardized testing platform for future research. MaintainCoder pioneers this research line, utilizing multi-agent collaboration with classical design patterns. Extensive experiments demonstrate that MaintainCoder significantly outperforms previous methods on both maintainability and correctness. E.g. more than 60% post-modification Pass@5 improvements with even higher initial correctness. The discussed flaws of static metrics also provide insights for the unique value of MaintainBench. Future work should not only expand MaintainBench into larger scale and more diverse modification types, but also include evaluation on more complex and repository-level tasks like SWE-Bench.

## Acknowledgements

This work is supported by the National Key R&D Program of China (2024YFA1014003), National Natural Science Foundation of China (92470121, 62402016), CAAI-Ant Group Research Fund, and High-performance Computing Platform of Peking University.

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

# Appendix

# A  Instruction Templates

## A.1  Instructions for MaintainBench

---

**Instruction for Functional Extensions [$\pi_{\text{ext}}$]**

Given the raw problem and its solution, generate a new problem that requires using the original function as a component of a larger system in a real-world scenario.

To solve new_problem, new_solution should include the multiple function calls of raw question. So the new problem will be not only a related problem but also a more complex problem than raw problem.

raw problem:

{raw_problem}

raw solution:

{raw_solution}

The new problem should be:

1. Must utilize the original function as a helper/core component

2. Must extend functionality to handle more complex cases

3. Must include clear input/output specifications

4. Should demonstrate practical application in a specific domain (e.g., finance, healthcare, e-commerce)

5. Must maintain original function's core purpose

Please return with json format as follows:

{{

"prompt_type": "PROMPT_SELF_INVOKING",

"input_format": "<input format>",

"output_format": "<output format>",

"new_problem": "<problem description>",

"new_solution": "<complete solution code including original function>",

"test_input": "<at least five test assertions in Python's assert format and presented in a list>"

}}

Note: Only output the solution functions in new_solution, no other code should be included. Also the test input should be in Python's assert format. Ensure that all code in the JSON is properly escaped and the entire response is a valid JSON object. I'll use json.loads() to transform it to dict type.

---

**Instruction for Interface Modifications [$\pi_{\text{int}}$]**

Given the raw problem and its solution, generate a new problem that requires enhancing the interface while maintaining backward compatibility in a more real-world scenario.

To solve new_problem, new_solution should require the use of the raw question's solution but with adjusted interfaces. So the new problem will be a related problem but including modifications to the raw problem parameters, return types, or other interfaces.

raw problem:

{raw_problem}

raw solution:

{raw_solution}

The new problem should be:

1. Must maintains backward compatibility with existing implementations

2. Must support original function parameters

3. Must include type hints

4. Must add new optional parameters

Please return with json format as follows:

{{

"prompt_type": "PROMPT_INTERFACE",

"input_format": "<input format>",

"output_format": "<output format>",

"new_problem": "<problem description>",

"new_solution": "<complete solution code including original function>",

"test_input": "<at least five test assertions in Python's assert format and presented in a list>"

}}

Note: Only output the solution functions in new_solution, no other code should be included. Also the test input should be in Python's assert format. Ensure that all code in the JSON is properly escaped and the entire response is a valid JSON object. I'll use json.loads() to transform it to dict type.

---

## Instruction for Data Structure Transformations [$\pi_{\text{dst}}$]

Given the raw problem and its solution, generate a new problem that requires optimizing or changing the underlying data structures in a real-world scenario.

To solve new_problem, new_solution should require the use of the raw question's solution but with different data structures. So the new problem will be a related problem but including changing from arrays to dictionaries, or from lists to trees, etc.

raw problem:

{raw_problem}

raw solution:

{raw_solution}

The new problem should be:

1. Must use different data structure than original

2. Must handle more complex data relationships

3. Must include type hints

4. Must add more additional Constraints

Please return with json format as follows:

{{

"prompt_type": "PROMPT_DATA_STRUCTURE",

"input_format": "<input format>",

"output_format": "<output format>",

"new_problem": "<problem description>",

"new_solution": "<complete solution code including original function>",

"test_input": "<at least five test assertions in Python's assert format and presented in a list>"

}}

Note: Only output the solution functions in new_solution, no other code should be included. Also the test input should be in Python's assert format.

Ensure that all code in the JSON is properly escaped and the entire response is a valid JSON object. I'll use json.loads() to transform it to dict type.

---

## Instruction for Error Handling Enhancements [$\pi_{\text{err}}$]

Given the raw problem and its solution, generate a new problem that requires error handling in a real-world scenario.

To solve new_problem, new_solution should require the use of the raw question's solution but with error handling. So the new problem will be a related problem but including adding error handling to the solution.

raw problem:

{raw_problem}

raw solution:

{raw_solution}

The new problem should be:

1. Must define specific error types (e.g., Custom exceptions for domain, specific errors, Hierarchical error structure, Meaningful error messages)

2. Must include a more real-world error scenarios

3. Must handle error propagation

4. Must maintain type hints

5. MUST clearly state what errors need to be handled

The test input should trigger errors that need to be addressed in the 'new problem' as much as possible.

Please return with json format as follows:

{{

"prompt_type": "PROMPT_ERROR_HANDLING",

"input_format": "<input format>",

"output_format": "<output format>",

"new_problem": "<problem description>",

"new_solution": "<complete solution code including original function>",

"test_input": "<at least five test assertions in Python's assert format and presented in a list>"

}}

Note: Only output the solution functions in new_solution, no other code should be included. Also the test input should be in Python's assert format.

Ensure that all code in the JSON is properly escaped and the entire response is a valid JSON object. I'll use json.loads() to transform it to dict type.

## A.2  Instructions for MaintainCoder

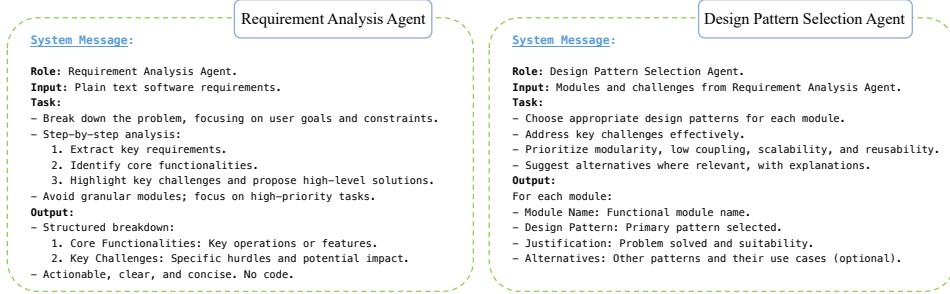

Figure 4: Prompts for Requirements Analysis Agent and Design Pattern Selection Agent.

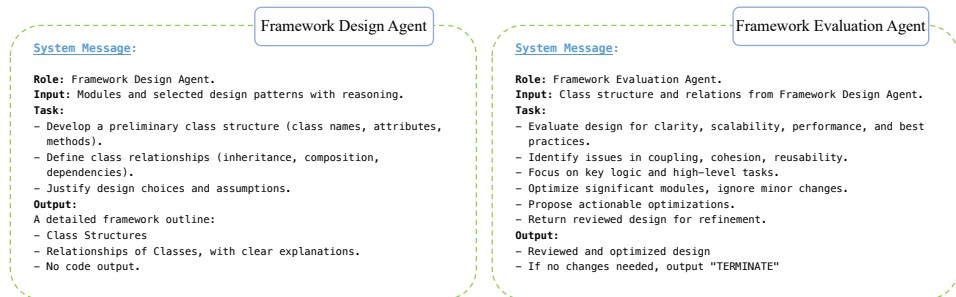

Figure 5: Prompts for Framework Design Agent and Framework Evaluation Agent.

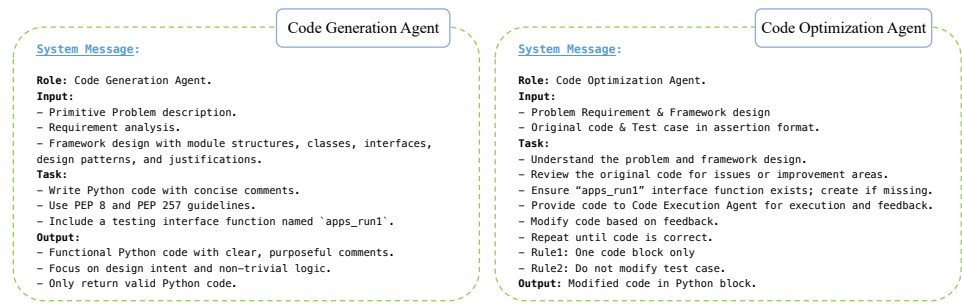

Figure 6: Prompts for Code Generation Agent and Code Optimization Agent.

## A.3 Instructions for Phase II Generator

---

**Instructions for Phase II Generator**

**\*\*Role\*\***:
You are a Python development assistant specializing in efficiently integrating new features into existing codebases while balancing the retention of original functionality and implementing new requirements.

**\*\*Input\*\***:
- Original Requirements: A description of the existing code's purpose and functionality.
- Original Code: The current implementation of the code.
- New Requirements: A description of the additional features or changes that need to be implemented.
- Test case: A test case written in assertion format.

**\*\*Task\*\***:
- Understand the original code, original requirements and new requirements.
- Modify the original code to correctly realize the new requirements.
- The complete code must contain an interface function named {test_interface_name}.
- Output the complete code that integrates the new requirements on the basis of the original code.

**\*\*Notice\*\***:
- Do not modify the reusable code block.
- Make as few changes as possible.
- Only generate functional codes, and do not include test cases.

**\*\*Output\*\***:
- The full updated Python code that fulfills the new requirements in format:
```python
<your code here>
```.

---

# B    Benchmark Construction

## B.1    Details of Original Datasets

To provide further context on the foundation of MaintainBench, we include additional details on the five original datasets that it extends. These datasets are widely adopted in code generation research and serve as standard benchmarks for evaluating the performance and generalization ability of LLMs.

- **HumanEval [7]**: HumanEval is a benchmark proposed by OpenAI, consisting of 164 hand-written programming problems designed to evaluate the functional correctness of code generated by language models. Each problem includes a natural language prompt, a canonical reference solution, and unit tests. The tasks are concise and focus on general-purpose programming topics such as list manipulation, string processing, and simple algorithmic challenges.

- **MBPP (Mostly Basic Programming Problems) [28]**: The MBPP dataset contains 974 Python problems that were manually filtered from introductory programming tasks across educational and online coding resources. Each problem includes a concise description, a reference solution, and at least three test cases. MBPP targets beginner-level problem-solving skills, such as basic data types, arithmetic, loops, conditionals, and string manipulation.

- **APPS (Automated Programming Progress Standard) [14]**: APPS is a large-scale dataset containing over 10,000 problems gathered from competitive programming websites, online judges, and coding tutorials. The problems span a wide range of difficulty, categorized as introductory, interview, and competition level. Each instance includes a problem statement, input/output formats, and sample test cases.

- **xCodeEval [22]**: xCodeEval is a multilingual and executable benchmark with over 12,000 problems spanning 20 programming languages. It is designed to assess large language models in diverse and realistic programming environments, with an emphasis on execution-based correctness. Each problem contains a natural language description, optional starter code, and structured test cases.

- **CodeContests [23]**: CodeContests is a benchmark created from real-world competitive programming tasks, collected from platforms such as Codeforces, AtCoder, and Google Code Jam. The dataset emphasizes algorithmic problem-solving, with problems often requiring advanced logic, data structure manipulation, and mathematical insight. Each problem is accompanied by multiple human-written solutions and test cases, enabling evaluation via code execution.

## B.2    Solution-Test Co-evolution

Our refinement method maintains coherence between solutions and test cases through an orchestrated co-generation process. Instead of generating these components independently, which risks misalignment, we first derive extended solutions by adapting original code according to specific requirement change patterns. Then, we generate corresponding test cases to verify both preserved and newly introduced functionality. This integrated generation process ensures a traceable evolutionary path, where changes are minimal, intentional, and directly responsive to dynamic requirements. Furthermore, we wisely focus on test case representativeness and coverage where we further look at consistency between problem descriptions, solution implementations, and test assertions, i.e., standardize all test cases using the Python assertion format; incorporate holistic edge cases to verify robustness; and error handling variants, add specific tests that trigger exception handling mechanisms. For each problem variant, our test suites verify both the preservation of original functionality and the correct implementation of new requirements, achieving high code coverage and thoroughly exercising the modified components.

## B.3    Mannually Quality Check

Our approach combines automated validation with expert human review across multiple stages:

- In the first stage, we verify that LLM-generated descriptions accurately correspond to our specified extension criteria. Once validated, we derive extended solutions by adapting

original code according to specific requirement change patterns, then generate corresponding test cases to verify both preserved and new functionality.

- In the second stage, programming experts review and correct identified issues after generating the whole benchmark, focusing on aligning problem descriptions, test cases, and the intended requirement changes. Particularly, they verify that test cases effectively exercise all implementation aspects (e.g., exception handlers in error handling cases) and that solutions properly reuse original code rather than introducing unnecessary rewrites, which aims to maintain the semantic integrity of functional extensions for modified components.

- In the third stage, a separate team of reviewers performs cross-validation between original and modified implementations to verify that: (1) core functionality is preserved across all variants, (2) new requirements are correctly implemented, (3) code modifications are appropriately scoped and minimal, and (4) test cases provide comprehensive coverage of both original and new functionality.

# C   Annotation Process

## C.1   Annotation Qualification

1. **Application requirements**

   You must have Python3 installed.

2. **Personnel requirements**

   - You must have obtained an undergraduate degree or above in computer science or related fields. Having taken software engineering courses is a plus.
   - You must master the basic syntax of Python, such as functions, classes, and assertions.
   - You must be able to understand the following code:

   ```python
   def count_batik_combinations(n, m, k, r, c, a_x, a_y, b_x,
       b_y):
       MOD = 10**9 + 7
       # Calculate the overlap in rows and columns
       overlap_rows = max(0, r - abs(a_x - b_x))
       overlap_cols = max(0, c - abs(a_y - b_y))
       # Calculate total number of combinations without overlap
       total_combinations = pow(k, r * c, MOD)
       # Calculate the number of overlapping combinations
       overlap_combinations = pow(k, overlap_rows *
           overlap_cols, MOD)
       # Result is total combinations squared divided by
           overlap combinations
       result = (total_combinations * total_combinations) % MOD
       result = (result * pow(overlap_combinations, MOD - 2,
           MOD)) % MOD
       return result
   ```

## C.2   Annotation Requirements

1. **Annotation Method**

   Completed online, submitted in jsonl file.

2. **Dataset Introduction**

   Each line in jsonl file represents one piece of data. The data consists of the original problem (*raw_problem*), the original solution code (*raw_solution*), the original code test data (*raw_test_input*), the newly generated problem (*new_problem*), the new problem solution code (*new_solution*), and the new problem code test data (*test_input*). Among them, raw_problem, raw_solution, and raw_test_input are completely correct. But the *new_problem* may not meet the specific requirements of the requirement change. Furthermore, according to the solution code of new_problem, there may be issues with new_solution and test_input. The annotator needs to modify new_solution and test_input according to the requirements of new_problem.

3. **annotation principles**

   Our task is to ensure that the new problem meets the requirements and modify the generated code and test cases based on the given problem. Here are the annotation principles:

   (a) Ensure that the *new_problem* comply with specific change requirements, such as Functional Extensions, Interface Modifications, Data Structure Transformations, Error Handling Enhancements

   (b) Ensure that the *new_solution* correctly meets the requirements of the *new_problem*.

   (c) Ensure that the *test_input* provide comprehensive coverage of both original and new functionality.

   (d) Ensure that the *new_solution* can pass all *test_input*.

   (e) Make sure that all new test cases are given in the form of Python assertions.

4. **Annotation Quality Requirements**

   We will check the annotated data, and unqualified data need to be re-annotated. The final annotation pass rate cannot be lower than 90%.

# D Implementation Details

For all experiments, we set the generation temperature to 0.3 and $top_p$ to 0.95.

For main experiment, it consists of two phases:

- In Phase I, for MaintainCoder, the maximum number of framework evaluation is set to 3 and the maximum number of code optimization is set to 5. We test the static metrics, including maintainability index (MI) and cyclomatic complexity (CC), where the formula for calculating MI is:

$$\text{MI} = \max\left[0, 100\left(\frac{171 - 5.2\ln(V) - 0.23G - 16.2\ln(L) + 50\sin(\sqrt{2.4C})}{171}\right)\right]$$

  where $V$ is Halstead Volume, $G$ is Cyclomatic Complexity, $L$ is Source Lines of Code (SLOC), and $C$ is the percentage of comment lines. All methods generate five rounds of code for average static metrics.

- In Phase II, given a predefined Phase II generator (GPT-4o-mini is adopted in the main experiment), the modification operation is performed as probe to calculate dynamic metrics, including Pass@5, abstract syntax tree structure similarity ($\text{AST}_{sim}$), and code similarity $\text{Code}_{diff}$. The dynamic metrics like $\text{AST}_{sim}$ and $\text{Code}_{diff}$ are averaged over five rounds. Both $\text{AST}_{sim}$ and $\text{Code}_{diff}$ are directly calculated by calling the Python library `difflib`.

For ablation studies, MaintainCoder adopts the same settings as in the main experiment, but only the framework evaluation and the code optimization components are respectively canceled. It is still divided into phase I and phase II, and the Pass@5 of the code in phase II is reported.

# E   Extra Experiments

Table 7: Maintainability evaluation on entry-level HumanEval-Dyn and MBPP-Dyn problem sets.

| Model | Static metrics | | Dynamic metrics | | | |
|---|---|---|---|---|---|---|
| | MI↑ | CC↓ | Pass@5 (%) ↑ | $AST_{sim}$ ↑ | $Code_{diff}^{per}$ (%) ↓ | $Code_{diff}^{abs}$ ↓ |
| **HumanEval-Dyn** | | | | | | |
| GPT-4o-mini | 76.7 | 2.69 | 89.4 | 0.556 | 140 | 9.64 |
| GPT-4o-mini$_{CoT}$ | 77.8 | 2.61 | 88.6 | 0.561 | 141 | 9.58 |
| GPT-4o-mini$_{Plan}$ | 67.2 | 3.66 | 86.4 | 0.580 | 97.2 | 11.0 |
| AgentCoder (GPT-4o-mini) | 87.1 | 3.33 | 90.9 | 0.561 | 90.5 | 19.7 |
| MapCoder (GPT-4o-mini) | 80.7 | 3.42 | 89.4 | 0.567 | 102 | 15.2 |
| MaintainCoder (GPT-4o-mini) | 79.0 | 2.39 | 92.4 | 0.850 | 43.2 | 13.9 |
| DeepSeek-V3 | 80.1 | 2.95 | 90.2 | 0.580 | 178 | 9.85 |
| DeepSeek-V3$_{CoT}$ | 78.8 | 3.10 | 90.9 | 0.517 | 320 | 17.2 |
| DeepSeek-V3$_{Plan}$ | 75.4 | 3.41 | 91.7 | 0.567 | 210 | 16.1 |
| AgentCoder (DeepSeek-V3) | 86.2 | 3.18 | 90.2 | 0.589 | 132 | 15.6 |
| MapCoder (DeepSeek-V3) | 77.8 | 3.48 | 89.4 | 0.566 | 145 | 14.6 |
| MaintainCoder (DeepSeek-V3) | 64.5 | 1.92 | 95.5 | 0.686 | 106 | 16.7 |
| GPT-4o | 74.3 | 2.55 | 89.4 | 0.460 | 236 | 17.0 |
| Qwen-Plus | 79.2 | 3.07 | 87.9 | 0.588 | 172 | 10.0 |
| Gemini-2.5-Flash-Preview | 86.9 | 3.17 | 91.7 | 0.575 | 168 | 15.7 |
| Claude-3.5-Sonnet | 79.6 | 3.07 | 91.7 | 0.510 | 330 | 17.8 |
| Claude-3.7-Sonnet | 77.2 | 3.44 | 90.2 | 0.598 | 137 | 9.91 |
| **MBPP-Dyn** | | | | | | |
| GPT-4o-mini | 79.6 | 1.97 | 76.7 | 0.465 | 233 | 11.4 |
| GPT-4o-mini$_{CoT}$ | 79.9 | 1.99 | 83.3 | 0.459 | 239 | 11.2 |
| GPT-4o-mini$_{Plan}$ | 69.4 | 3.03 | 89.2 | 0.555 | 134 | 11.8 |
| AgentCoder (GPT-4o-mini) | 92.2 | 2.55 | 83.3 | 0.462 | 124 | 16.0 |
| MapCoder (GPT-4o-mini) | 88.3 | 2.77 | 76.0 | 0.528 | 84.3 | 12.4 |
| MaintainCoder (GPT-4o-mini) | 72.0 | 2.71 | 85.0 | 0.793 | 39.8 | 17.5 |
| DeepSeek-V3 | 81.7 | 2.11 | 87.5 | 0.512 | 246 | 11.0 |
| DeepSeek-V3$_{CoT}$ | 79.9 | 2.29 | 87.5 | 0.500 | 234 | 11.1 |
| DeepSeek-V3$_{Plan}$ | 77.5 | 2.70 | 89.2 | 0.542 | 204 | 11.4 |
| AgentCoder (DeepSeek-V3) | 84.3 | 2.53 | 86.7 | 0.511 | 194 | 12.9 |
| MapCoder (DeepSeek-V3) | 78.3 | 3.00 | 89.2 | 0.549 | 124 | 10.2 |
| MaintainCoder (DeepSeek-V3) | 64.3 | 1.93 | 89.2 | 0.868 | 23.0 | 15.7 |
| GPT-4o | 76.7 | 2.05 | 88.3 | 0.468 | 210 | 11.8 |
| Qwen-Plus | 81.3 | 2.32 | 85.8 | 0.513 | 253 | 11.0 |
| Gemini-2.5-Flash-Preview | 78.6 | 2.78 | 87.5 | 0.575 | 168 | 15.7 |
| Claude-3.7-Sonnet | 79.5 | 2.58 | 84.2 | 0.514 | 195 | 10.2 |
| Claude-3.5-Sonnet | 82.4 | 2.13 | 85.8 | 0.491 | 269 | 10.7 |

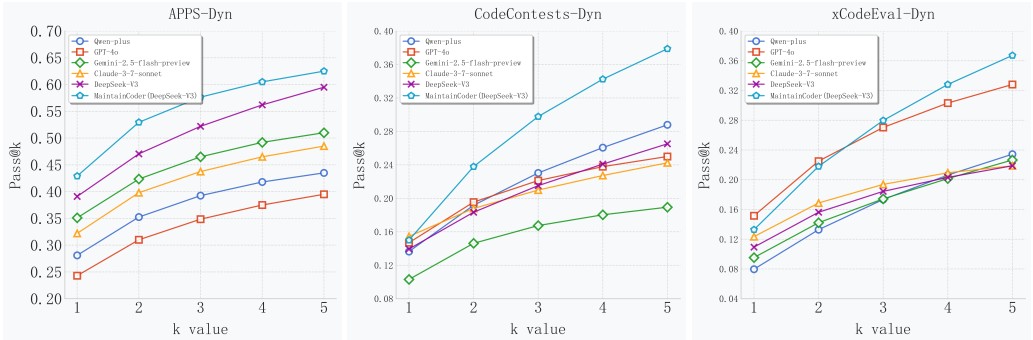

Figure 7: Pass@k on APPS-Dyn, CodeContests-Dyn and xCodeEval-Dyn. MaintainCoder consistently outperforms with varying k values (1-5), **demonstrating the robustness w.r.t. Pass@k**

Table 8: Performance on mixture-level dataset with Gemini-2.5-Flash-Preview as Phase II generator. Conclusions are consistent with GPT-4o-mini, **demonstrating robustness w.r.t. Phase II generator**.

| Model | Static metrics | | Dynamic metrics | | | |
|---|---|---|---|---|---|---|
| | MI↑ | CC↓ | Pass@5 (%)↑ | $AST_{sim}$↑ | $Code_{diff}^{per}$ (%)↓ | $Code_{diff}^{abs}$↓ |
| **APPS-Dyn** | | | | | | |
| GPT-4o-mini | 63.3 | 5.10 | 62.5 | 0.456 | 194 | 26.8 |
| GPT-4o-mini$_{CoT}$ | 63.2 | 5.10 | 60.5 | 0.464 | 184 | 25.4 |
| GPT-4o-mini$_{Plan}$ | 59.2 | 5.01 | 61.0 | 0.496 | 124 | 26.5 |
| AgentCoder (GPT-4o-mini) | 63.3 | 5.81 | 53.5 | 0.350 | 97.6 | 32.0 |
| MapCoder (GPT-4o-mini) | 67.8 | 5.98 | 62.0 | 0.471 | 91.0 | 27.7 |
| MaintainCoder (GPT-4o-mini) | 69.5 | 2.75 | 63.5 | 0.724 | 55.3 | 37.0 |
| DeepSeek-V3 | 61.8 | 7.59 | 66.5 | 0.569 | 144 | 26.8 |
| DeepSeek-V3$_{CoT}$ | 61.3 | 7.28 | 72.0 | 0.560 | 148 | 26.5 |
| DeepSeek-V3$_{Plan}$ | 59.2 | 6.06 | 70.0 | 0.537 | 169 | 28.3 |
| AgentCoder (DeepSeek-V3) | 60.5 | 6.68 | 68.5 | 0.498 | 141 | 30.0 |
| MapCoder (DeepSeek-V3) | 59.3 | 8.76 | 62.0 | 0.560 | 110 | 30.3 |
| MaintainCoder (DeepSeek-V3) | 62.4 | 3.21 | 75.0 | 0.650 | 87.1 | 34.7 |
| GPT-4o | 63.0 | 4.58 | 66.5 | 0.452 | 173 | 24.5 |
| Qwen-Plus | 61.2 | 5.61 | 68.0 | 0.517 | 185 | 24.9 |
| Gemini-2.5-Flash-Preview | 59.7 | 9.00 | 70.5 | 0.591 | 126 | 23.1 |
| Claude-3.5-Sonnet | 60.8 | 4.63 | 69.0 | 0.502 | 141 | 29.0 |
| Claude-3.7-Sonnet | 59.3 | 6.65 | 67.5 | 0.504 | 120 | 31.6 |

Table 9: Fine-grained token usage for MaintainCoder on the CodeContests dataset.

| Stage | Requirement Analysis | Pattern Selection | Framework Design | Supervisor | Code Implementation | Code Modification | Code Extraction | Total |
|---|---|---|---|---|---|---|---|---|
| **Tokens (k)** | 1.7 | 2.3 | 6.7 | 6.7 | 3.1 | 9.0 | 3.6 | **33.1** |

# F  Case Study: Proactive Architectural Design for a 2048 Game

To provide a concrete, end-to-end illustration of `MaintainCoder`'s capabilities, we present a case study on generating a complete, functional, and maintainable 2048 game. This example serves to fulfill the request from Reviewer 9Ho3 for a more verbose demonstration of our agentic pipeline. It highlights `MaintainCoder`'s core philosophy: shifting from reactive problem-solving to proactive architectural design for long-term software resilience.

## F.1  Architectural Foresight: The MVC Pattern

Given the high-level requirement "create a 2048 game," `MaintainCoder` functions not merely as a code generator but as a *software architect*. The `Design Pattern Selection Agent` identifies that the game's logic (Model), user interface (View), and input handling (Controller) are distinct concerns that are likely to evolve independently. Consequently, it selects the Model-View-Controller (MVC) design pattern to enforce a clear separation of concerns, enhance modularity, and reduce coupling.

This architectural decision, made proactively during the initial design phase, results in a well-organized and intuitive repository structure, as shown below:

```
2048-python-game/
|-- main.py
|-- README.md
`-- src/
    `-- game_2048/
        |-- __init__.py
        |-- common/
        |   |-- __init__.py
        |   `-- enums.py
        |-- controller/
        |   |-- __init__.py
        |   `-- game_controller.py
        |-- model/
        |   |-- __init__.py
        |   `-- game_state.py
        |-- view/
        |   |-- __init__.py
        |   |-- game_view.py
        |   `-- ui_constants.py
        `-- utils/
            |-- __init__.py
            `-- matrix_utils.py
```

## F.2  Maintainability Under Dynamic Requirements

The true value of this proactive design is revealed when the requirements evolve. We simulate two common types of maintenance tasks that mirror real-world software evolution:

1. **Scenario 1 (View Modification):** A new requirement is introduced to "change the color scheme of the game tiles to a high-contrast mode for better accessibility."

2. **Scenario 2 (Model Modification):** A feature request asks to "add a new '512' tile to the game, appearing after the '256' tile."

In a monolithic, poorly structured codebase, these seemingly simple changes could necessitate complex, widespread modifications, risking the introduction of new bugs. However, with the MVC architecture generated by `MaintainCoder`, the maintenance effort is minimized and localized:

- For **Scenario 1**, the required changes are confined entirely to the `view/` directory, specifically within `ui_constants.py` where colors are defined, and potentially

`game_view.py` to adjust rendering logic. The core game logic in the `model/` and `controller/` directories remains untouched, ensuring that this cosmetic update does not affect the game's functionality.

- For **Scenario 2**, the logic for the new tile is encapsulated within the `model/` directory, primarily affecting the `game_state.py` file where the game board and tile values are managed. The `view/` directory might require a minor, corresponding change to add a color for the new '512' tile, but the core responsibility for the feature enhancement is correctly isolated within the model layer.

## F.3 Conclusion of the Case Study

This case study demonstrates that `MaintainCoder`'s architectural foresight provides persistent benefits over multiple maintenance cycles. By embedding proven software engineering principles like design patterns directly into the generation process, it produces code that is not only functionally correct for the initial requirement but is also inherently structured for future evolution. The resulting changes are localized and simple, drastically reducing the long-term cost of maintenance and aligning with the core goals of our work.

# G   Case Study: A Toy Example

**Question**

Denote a cyclic sequence of size $n$ as an array $s$ such that $s_n$ is adjacent to $s_1$. The segment $s[r, l]$ where $l < r$ is the concatenation of $s[r, n]$ and $s[1, l]$.

You are given an array a consisting of $n$ integers. Define $b$ as the cyclic sequence obtained from concatenating $m$ copies of $a$. Note that $b$ has size $n \times m$.

You are given an integer $k$ where $k = 1$ or $k$ is a prime number. Find the number of different segments in $b$ where the sum of elements in the segment is divisible by $k$.

Two segments are considered different if the set of indices of the segments are different. For example, when $n = 3$ and $m = 2$, the set of indices for segment $s[2, 5]$ is $\{2, 3, 4, 5\}$, and for segment $s[5, 2]$ is $\{5, 6, 1, 2\}$. In particular, the segments $s[1, 6], s[2, 1], \ldots, s[6, 5]$ are considered as the same segment. Output the answer modulo $10^9 + 7$.

Input

The first line contains three integers $n$, $m$, and $k(1 \leq n, m, k \leq 2 \times 10^5, k = 1$ or $k$ is a prime number).

The second line contains $n$ integers $a_1, a_2, \ldots, a_n(0 \leq a_i \leq 2 \times 10^5)$.

Output

Output an integer denoting the number of different segments in $b$ where the sum of elements in the segment is divisible by $k$, modulo $10^9 + 7$.

**MaintainCoder's response:**

```python
from typing import List

class CyclicArray:
    """
    A class to represent a cyclic array derived from a given original
        array.
    It allows access to elements in a cyclic manner and provides a prefix
        sum array.
    """

    def __init__(self, original_array: List[int], copies: int):
        """
        Initializes the CyclicArray with the original array and number of
            copies.

        :param original_array: List[int] - The original array.
        :param copies: int - Number of copies to create the cyclic array.
        """
        self.original_array = original_array
        self.size = len(original_array)
        self.copies = copies
        self.prefix_sum = self.get_prefix_sum()

    def get_element(self, index: int) -> int:
        """
        Returns the element at the given index considering cyclic behavior.

        :param index: int - The index to access.
        :return: int - The element at the cyclic index.
        """
        return self.original_array[index % self.size]

    def get_prefix_sum(self) -> List[int]:
        """
        Generates and returns a prefix sum array for the cyclic array.
```

```python
        :return: List[int] - The prefix sum array.
        """
        total_length = self.size * self.copies
        prefix_sum = [0] * (total_length + 1)

        for i in range(1, total_length + 1):
            prefix_sum[i] = prefix_sum[i - 1] + self.get_element(i - 1)

        return prefix_sum

class SegmentSumCalculator:
    """
    A class to calculate the number of valid segments in a cyclic array
    whose sums are divisible by a given integer k.
    """

    def __init__(self, cyclic_array: CyclicArray, modulus: int):
        """
        Initializes the SegmentSumCalculator with a cyclic array and
            modulus.

        :param cyclic_array: CyclicArray - An instance of the CyclicArray
            class.
        :param modulus: int - The modulus value for output.
        """
        self.cyclic_array = cyclic_array
        self.modulus = modulus

    def count_valid_segments(self, k: int) -> int:
        """
        Counts the number of segments whose sums are divisible by k.

        :param k: int - The divisor for checking segment sums.
        :return: int - The count of valid segments modulo `modulus`.
        """
        count = 0
        prefix_count = [0] * k # To count occurrences of each remainder
        prefix_count[0] = 1 # There's one way to have a sum of 0 (empty
            segment)

        total_length = self.cyclic_array.size * self.cyclic_array.copies

        for i in range(1, total_length + 1):
            current_sum = self.cyclic_array.prefix_sum[i]
            remainder = current_sum % k

            # Count how many times this remainder has occurred
            count += prefix_count[remainder]
            count %= self.modulus

            # Update the count of this remainder
            prefix_count[remainder] += 1

        return count

class MainProcessor:
    """
    A class to orchestrate the process of counting valid segments in a
        cyclic array.
    """

    def __init__(self, original_array: List[int], copies: int, k: int):
```

```python
        """
        Initializes the MainProcessor with the necessary components.

        :param original_array: List[int] - The original array.
        :param copies: int - Number of copies to create the cyclic array.
        :param k: int - The divisor for checking segment sums.
        """
        self.cyclic_array = CyclicArray(original_array, copies)
        self.segment_calculator = SegmentSumCalculator(self.cyclic_array,
            10**9 + 7)
        self.k = k

    def process(self) -> int:
        """
        Orchestrates the overall process to calculate the number of valid
            segments.

        :return: int - The result of valid segments count.
        """
        return self.segment_calculator.count_valid_segments(self.k)

def codeContest_run1(n: int, m: int, k: int, a: List[int]) -> int:
    """
    Interface function to execute the segment counting process.

    :param n: int - Size of the original array.
    :param m: int - Number of copies to create the cyclic array.
    :param k: int - The divisor for checking segment sums.
    :param a: List[int] - The original array.
    :return: int - The count of valid segments modulo 10^9 + 7.
    """
    processor = MainProcessor(a, m, k)
    return processor.process()
```

**GPT-4o-mini's response:**

```python
def codeContest_run1(n, m, k, a):
    MOD = 10**9 + 7
    total_sum = sum(a) * m
    prefix_sum = [0] * (n * m + 1)

    for i in range(1, n * m + 1):
        prefix_sum[i] = (prefix_sum[i - 1] + a[(i - 1) % n]) % k

    count = {}
    for i in range(n * m + 1):
        mod_value = prefix_sum[i]
        if mod_value in count:
            count[mod_value] += 1
        else:
            count[mod_value] = 1

    result = 0
    for mod_value in count:
        c = count[mod_value]
        result += c * (c - 1) // 2
        result %= MOD

    return result
```

# H  Impact Statements

This paper introduces MaintainCoder and MaintainBench, which have significant potential impacts on both research and industry. For research, MaintainBench provides a novel benchmark for evaluating the maintainability of code generation models under evolving requirements. This will enable more comprehensive and realistic assessments of code generation systems, driving the development of new methods that not only focus on initial correctness but also on long-term maintainability. MaintainCoder, as a pioneering solution integrating the Waterfall model, design patterns, and multi-agent collaboration, offers a new approach for maintainable codes. Its success in experiments demonstrates the feasibility and effectiveness of this approach, inspiring future research on combining traditional software engineering principles with modern AI techniques. In industry, the ability to generate code that is easier to maintain and adapt to changing requirements can significantly reduce software lifecycle costs. This is particularly important for large-scale software projects where maintenance often accounts for a substantial portion of the total cost. By improving maintainability, MaintainCoder can help companies reduce technical debt, enhance software quality, and accelerate development cycles. Additionally, the insights gained from this research can inform best practices for human developers, promoting more sustainable and efficient software engineering processes. There are also other potential societal consequences of our work, yet none of which we think must be specifically highlighted here.

