# OpenReview forum: "MaintainCoder: Maintainable Code Generation Under Dynamic Requirements"
_NeurIPS.cc/2025/Conference — NeurIPS 2025 poster_

### Official Review · Reviewer_wgB7 · 2025-06-29

**Clarity:** 3
**Significance:** 3
**Originality:** 2
**Rating:** 4
**Confidence:** 3

**Summary:**

This paper addresses a critical gap in code generation by focusing on maintainability under evolving requirements rather than just initial correctness. The authors introduce two main contributions: (1) **MaintainBench**, the first benchmark for evaluating code maintainability through requirement evolution cycles, extending five established datasets (HumanEval, MBPP, APPS, CodeContests, xCodeEval) with systematic requirement changes; and (2) **MaintainCoder**, a multi-agent system that integrates the Waterfall model, design patterns, and multi-agent collaboration to generate maintainable code. The work proposes novel dynamic metrics (Pass@k, ASTsim, Codediff) to evaluate maintenance efforts.

**Questions:**

- Given the significant reliance on GPT-4o for MaintainBench construction, how do you ensure benchmark validity and avoid favoring GPT-based systems? Could you provide inter-annotator agreement scores and more details on the manual validation process to demonstrate the quality of human oversight?
- Could you provide detailed computational overhead analysis comparing MaintainCoder to baseline methods?
- How would MaintainCoder perform across multiple sequential requirement changes rather than single cycles? Have you considered evaluating cumulative effects of iterative maintenance on code quality, which would better reflect real-world software evolution?

**Ethical Concerns:**

["NO or VERY MINOR ethics concerns only"]

**Final Justification:**

The author's rebuttal addresses some of the concerns. However, the comparison with previous works such as MetaGPT and EvoMAC is inconsistent in terms of LLM settings, alternating between gpt-4o-mini and gpt-4.1-mini. Additionally, the claimed novelty of the Reactive vs. Proactive distinction does not account for EvoMAC, which also follows a proactive evolution approach. Overall, while the paper still requires further improvements, it has its merits and contributions. I will maintain a positive score.

**Limitations:**

Yes

**Paper Formatting Concerns:**

Texts in Figure 3 is too small.

**Quality:**

2

**Strengths And Weaknesses:**

### Strengths

- The paper addresses a critical and underexplored gap in code generation by focusing on maintainability under evolving requirements rather than just initial correctness.
- MaintainBench represents a novel and valuable contribution as the first benchmark for evaluating dynamic maintainability through requirement evolution cycles, with systematic extension of five established datasets and four well-motivated requirement change patterns.
- The work provides both immediate practical value and opens promising research directions, potentially influencing how the community evaluates code generation systems.

### Weaknesses

- The mathematical formulation, this is, the Monte Carlo approximation with first-order truncation may oversimplify the complexity of long-term maintenance scenarios. Need more explanation about the motivation and relationship between the formulation and the real-world unlimited requirement evolution scenarios. It would make this idea more clear by going through a real-world coding case and the examples about each notation in this dynamic evolving software development case.
- MaintainCoder's design lacks novelty. It applies a fixed conventional software development workflow (Waterfall model + design patterns) within a multi-agent framework. This type of workflow-fixed multi-agent system mimicking real development scenarios has been extensively discussed and explored in prior work[1,2,3]. Could you compare with these methods and highlight your novelty? Why this framework could handle the dynamic requirements while others could not?
- Given that maintainability principles should be language-agnostic, MaintainBench's limitation to Python represents weakness where the benchmark should ideally support different statically-typed languages and diverse programming paradigms. Could this method be extend to more general language scenarios?

[1] Hong S, Zheng X, Chen J, et al. Metagpt: Meta programming for multi-agent collaborative framework[J]. arXiv preprint arXiv:2308.00352, 2023, 3(4): 6.

[2] Qian C, Liu W, Liu H, et al. Chatdev: Communicative agents for software development[J]. arXiv preprint arXiv:2307.07924, 2023.

[3] Hu Y, Cai Y, Du Y, et al. Self-evolving multi-agent collaboration networks for software development[J]. arXiv preprint arXiv:2410.16946, 2024.

---

> ### Author Rebuttal · Authors · 2025-07-30
>
> Dear Reviewer wgB7:
>
> We sincerely appreciate your valuable feedback and suggestions, which are helpful in improving our paper.
>
> Below, we address each comment in detail.
>
> ## Response to W1:
>
> > **(Long-term Maintenance)** Monte Carlo approximation with first-order truncation may oversimplify the complexity of long-term maintenance scenarios. Need more explanation about the motivation and relationship between the formulation and the real-world unlimited requirement evolution scenarios. A real-world coding case would make this idea more clear.
>
> We agree that real-world maintenance is a long-term, multi-round process. However, simulating multi-round maintenance for evaluation is computationally prohibitive. We believe that **code maintainability is an intrinsic property**, and thus the single-round maintenance cost should be highly correlated with the cumulative cost of multiple rounds. **A single round, therefore, can serve as an effective and efficient proxy.**
>
> To validate this, we synthesized a second round of requirement changes for MaintainBench. As shown below, the costs from the **second round** **exhibit a high correlation with the first-round costs**, and are slightly more than the first ones. This suggests that the first-order approximation provides a good balance between evaluation accuracy and computational feasibility.
>
> *Tab1: First-round and Second-round Maintenance Costs (AST_sim & Code_diff) on CodeContests-Dyn.*
>
> | Model  | 1st Round Cost |  | 2nd Round Cost |  |
> |  - |  - |  - |  - |  - |
> |  | AST_sim ↑  | Code_diff (%) ↓ | AST_sim ↑  | Code_diff (%) ↓ |
> | GPT-4o-mini  | 0.661  | 90.1  | 0.645  | 96.2  |
> | MaintainCoder (GPT-4o-mini) | 0.833  | 23.2  | 0.811  | 29.4  |
> | DeepSeek-V3  | 0.718  | 87.2  | 0.694  | 92.5  |
> | MaintainCoder (DeepSeek-V3) | 0.788  | 43.2  | 0.765  | 48.9  |
>
> Given word count limit, we provide 2048 game for a coding case.
>
> ```Plain
> 2048-python-game/
> ├── README.md
> ├── main.py
> └── src/
>   └── game_2048/
>   ├── __init__.py
>   │
>   ├── common/
>   │  ├── __init__.py
>   │  └── enums.py
>   │
>   ├── controller/
>   │  ├── __init__.py
>   │  └── game_controller.py
>   │
>   ├── model/
>   │  ├── __init__.py
>   │  └── game_state.py
>   │
>   ├── view/
>   │  ├── __init__.py
>   │  ├── game_view.py
>   │  └── ui_constants.py
>   │
>   └── utils/
>   ├── __init__.py
>   └── matrix_utils.py
> ```
>
> MaintainCoder proactively adopt an MVC (Model-View-Controller) architecture. This separation of concerns is an intrinsic quality of the code. If the first requirement is "change the color of the tiles" (a View modification), and the second is "add a new '512' tile" (a Model modification), the changes are localized and simple. **This architectural foresight provides persistent benefits over multiple maintenance cycles.**
>
> We will add the discussions in the revision.
>
> ## Response to W2:
>
> > **(Design Novelty)** Could you compare with prior work that mimics real development scenarios and highlight your novelty? Why could this framework handle the dynamic requirements while others could not?
>
> The fundamental distinction and novelty of our work lie in a paradigm shift: from reactive problem-solving to proactive architectural design.
>
> - **Reactive vs. Proactive:** Prior systems like MetaGPT and ChatDev [1, 2] are *reactive*; they are optimized to produce a functionally correct solution for the *current* requirements. In contrast, MaintainCoder is *proactive*; it is designed to generate a solution that is not only correct now but also robust and easy to adapt.
> - **Focus on Architectural Quality:** Our framework's novelty is not merely in its multi-agent structure but in **explicitly embedding proven** **software engineering** **principles** (e.g., design patterns, separation of concerns) into the agents' core logic. This ensures the code is designed for evolution, not just for initial generation.
>
> The supplemented experiments are below.  We will add these results and a detailed discussion of related works [1, 2, 3, 4] to better position our contribution.
>
> *Tab2: More multi-agent baselines (MetaGPT needs gpt-4.1-mini for strong tool call capability, ChatDev has different output format and is not listed).*
>
> | CodeContest-Dyn  | MI↑  | CC↓  | pass@5↑ | AST_sim↑ | Code_diff^per↓ | Code_diff^abs↓ |
> |  - |  - |  - |  - |  - |  - |  - |
> | MaintainCoder (gpt-4o-mini) | 65.8 | 2.68 | 32.6    | 0.833    | 23.2           | 17.4           |
> | MetaGPT (gpt-4.1-mini)  | 55.2 | 7.63 | 30.3  | 0.760  | 44.3  | 18.6  |
> | EvoMAC (gpt-4o-mini)  | 62.6 | 5.18 | 26.5  | 0.685  | 60.1  | 20.0  |
>
> ## Response to W3:
>
> > **(Language Restriction)** MaintainBench ideally should support different statically-typed languages and diverse programming paradigms. Could this method be extended to more general languages?
>
> Yes! For industry, design patterns are successful and widely adopted in Java/C++ software development. For experiments, we translated MaintainBench from Python to Java, and evaluated MaintainCoder against the baseline on Java version. The results are consistent with Python: **MaintainCoder improves maintainability metrics, indicating our principles are not language-specific**. The Java performance is generally slightly lower than Python, due to imbalanced training data. We will supplement these discussion in the revision.
>
> *Tab3: Java version performance on CodeContests-Dyn.*
>
> | Model  | MI ↑ | CC ↓ | Pass@5 (%) ↑ | AST_sim ↑ | Code_diff (%) ↓ | Code_diff (abs) ↓ |
> |  - |  - |  - |  - |  - |  - |  - |
> | GPT-4o-mini  | 56.5 | 6.52 | 22.5  | 0.640  | 95.3  | 21.2  |
> | MaintainCoder (GPT-4o-mini) | 64.1 | 3.11 | 30.5  | 0.812  | 28.4  | 20.5  |
> | DeepSeek-V3  | 54.2 | 7.21 | 25.0  | 0.703  | 91.8  | 23.8  |
> | MaintainCoder (DeepSeek-V3) | 62.5 | 3.84 | 36.0  | 0.795  | 45.1  | 22.7  |
>
> ## Response to Q1:
>
> > **(Benchmark Construction)** How do you ensure benchmark validity and avoid favoring GPT-based systems? Provide inter-annotator agreement scores and more details on the manual validation process to demonstrate the quality of human oversight.
>
> **Model Bias:** Despite GPT-4o was used for the initial generation, the benchmark validity ultimately relies on human expert validation rather than just the LLMs. **All final test cases and solutions underwent execution-based verification and expert review,** ensuring that the ground truth is robust to the generating model biases.
>
> **Annotation details:** Each new generated solution and test input were annotated through recruited annotator and independently validated by two of authors. To further assess annotation reliability, **we obtained inter-annotator agreement using Fleiss’ Kappa from two of our authors, obtaining a score of 0.899, indicating high reliability and vadility** [7]. These details will be included in the revision.
>
> ## Response to Q2:
>
> > **(Token Cost)** Could you provide detailed computational overhead analysis.
>
> *Tab4: Computational overhead analysis (token usage).*
>
> | **Token Usage** | **MaintainCoder** | **MapCoder** | **o3-mini** | **MetaGPT/ChatDev** | **GPT-4o-mini** |
> |  - |  - |  - |  - |  - |  - |
> | CodeContests  | 33.1k  | 38.7k  | 20.8k  | 50k+  | 2.5k  |
> | xCodeEval  | 29.6k  | 23.5k  | 21.2k  | 50k+  | 2.3k  |
>
> We have performed the requested analysis and find that multi-agent pipelines incur higher computational costs. But for initial code generation, we believe this is a **worthwhile trade-off**. Specifically, the token usage of MaintainCoder is on par with MapCoder or reasoning model o3-mini, and much lower than MetaGPT/ChatDev [5,6]. But **MaintainCoder achieves a >60% improvement in maintainability and higher correctness**. The initial investment is justified by the drastic reduction in subsequent human effort required for debugging and refactoring. Table below details the token consumption per stage in our pipeline.
>
> *Tab5: More fine-grained computational overhead analysis (token usage).*
>
> | CodeContests | Requirement Analysis | Design Pattern Selection | Framework Design | Supervisor | Code Implementation | Code Modification | Code Extraction | Total |
> |  - |  - |  - |  - |  - |  - |  - |  - |  - |
> | tokens(k)  | 1.7  | 2.3  | 6.7  | 6.7  | 3.1  | 9..0  | 3.6  | 33  |
>
> ## Response to Q3:
>
> > **(Long-term Maintenance)** How would MaintainCoder perform across multiple sequential requirement changes rather than single cycles? Have you considered evaluating cumulative effects of iterative maintenance on code quality, which would better reflect real-world software evolution?
>
> For multiple sequential requirement changes, please refer to **Response to W1**.
>
> We agree that the evaluation of "cumulative effects of iterative maintenance on code quality" is a valuable research direction. **Such a study would need to measure multiple facets of quality, including not only correctness and efficiency but also the very concept of maintainability that our work focuses on**. Indeed, a benchmark like MaintainBench would be a necessary component for maintainability measure. We will add this point to our "Future Work" section, highlighting it as a promising avenue for research on code generation for long-term software evolution.
>
> Thank you again for your thoughtful review and for helping improve our work. If you have any further questions or suggestions, please let us know at any time.
>
> ## Reference
>
> [1] Metagpt: Meta programming for multi-agent collaborative framework
>
> [2] Chatdev: Communicative agents for software development
>
> [3] Self-evolving multi-agent collaboration networks for software development
>
> [4] SWE-Dev: Evaluating and Training Autonomous Feature-Driven Software Development
>
> [5] MapCoder: Multi-Agent Code Generation for Competitive Problem Solving
>
> [6] AgentCoder: Multi-Agent-based Code Generation with Iterative Testing and Optimisation
>
> [7] Fleiss' Kappa: Measuring Agreement Among Multiple Raters

---

> ### Author Response · Authors · 2025-08-05
> **To Confirm Concerns Addressed & Thanks for Review**
>
> Dear Reviewer wgB7,
>
> As the discussion period will conclude soon, **we hope to confirm that your concerns have been addressed**.
>
> For your convenience, we have summarized our key responses below:
>
> - **Response to W1 (Long-term Maintenance):**
>
> We demonstrate that single-round maintenance is an effective and efficient proxy for long-term scenarios, as our experiments show a high correlation between first and second-round maintenance costs. Our proactive architectural design provides persistent benefits across multiple updates, justifying this first-order approximation.
>
> - **Response to W2 (Design Novelty):**
>
> Our core novelty is the paradigm shift from reactive problem-solving to proactive architectural design, which explicitly embeds software engineering principles to generate inherently robust and adaptable code. Supplemental experiments show MaintainCoder's superiority in handling dynamic requirements compared to other multi-agent systems (MaintainCoder > EvoMAC > MapCoder > AgentCoder).
>
> - **Response to W3 (Language Restriction):**
>
> We confirm our method's generalizability by extending MaintainBench to Java, where our approach again significantly improves maintainability, proving the underlying software design principles are not language-specific.
>
> - **Response to Q1 (Benchmark Construction):**
>
> The benchmark's validity is ensured by rigorous human expert oversight and execution-based verification, which mitigates potential model bias from data generation and achieves a high inter-annotator agreement (Fleiss' Kappa = 0.899).
>
> - **Response to Q2 (Token Cost):**
>
> The initial computational overhead is a justified trade-off, as it achieves a greater than 60\% improvement in maintainability while using comparable or even fewer tokens than other multi-agent frameworks like MapCoder and MetaGPT.
>
> - **Response to Q3 (Long-term Maintenance):**
>
> As shown in our response to W1, our method performs well across sequential requirement changes, and we agree that evaluating the cumulative effects on code quality is a valuable future research direction for which our work provides a foundational tool.
>
> We have spared no effort in addressing your concerns, and remain open to discussing any further questions or clarifications. Thank you again for improving our paper!
>
> Best wishes,
>
> The Authors

---

> > ### Comment · Reviewer_wgB7 · 2025-08-05
> > **Response**
> >
> > Thanks for the rebuttal. The author's rebuttal addresses some of the concerns. However, the comparison with previous works such as MetaGPT and EvoMAC is inconsistent in terms of LLM settings, alternating between gpt-4o-mini and gpt-4.1-mini. Additionally, the claimed novelty of the Reactive vs. Proactive distinction does not account for EvoMAC, which also follows a proactive evolution approach.

---

> ### Author Response · Authors · 2025-08-06
> **Further Response to W2**
>
> Dear Reviewer wgB7,
>
> We appreciate so much for your quick response, and are gald to hear that your concerns, with the exception of W2, have been addressed. Please allow us to provide further clarification on this point.
>
> First, the results presented in previous response are only supplements. **For your convenience, here we combine these with existing multi-agent results in our paper, and present a consistent comparison** using GPT-4o-mini.
>
> As shown below, **multi-agent systems focused on problem-solving do not necessarily improve code maintainability compared to the base model** (e.g., AgentCoder). However, because MaintainCoder explicitly incorporates the wisdom of long-term human engineering practices—classic design patterns—it achieves significant optimization in maintainability.
>
> | CodeContest-Dyn             | MI↑  | CC↓  | pass@5↑ | AST_sim↑ | Code_diff^per↓ |
> | --------------------------- | ---- | ---- | ------- | -------- | -------------- |
> | GPT-4o-mini                 | 57.8 | 6.06 | 24.2    | 0.661    | 90.1           |
> | MaintainCoder (GPT-4o-mini) | 65.8 | 2.68 | 32.6    | 0.833    | 23.2           |
> | EvoMAC (GPT-4o-mini)        | 62.6 | 5.18 | 26.5    | 0.685    | 60.1           |
> | AgentCoder (GPT-4o-mini)    | 62.5 | 7.28 | 18.2    | 0.629    | 44.9           |
>
>
> Second, it is worth noting that **EvoMAC also demonstrates an optimization effect on maintainability compared to the base model and other multi-agent systems** like AgentCoder. EvoMAC optimizes code maintainability through an evolutionary approach. Although it does not explicitly introduce design patterns, it achieves good results through multiple iterations and optimizations.
>
> However, MaintainCoder's explicit introduction of the concept of design patterns allows it to optimize code maintainability more directly, efficiently, and with greater focus. **To draw an analogy, EvoMAC's evolutionary approach is like finding the optimal solution through continuous trial and error, whereas MaintainCoder is like directly applying a proven, efficient algorithm to solve the problem**. While EvoMAC's textual backpropagation is novel and evolvable, it is focused on solving the current requirements via evolution.
>
> Finally, for the implementation of the relevant multi-agent systems, we adopted the official code repositories and execution scripts for fairness. We have attempted to use MetaGPT with GPT-4o-mini, but we found that because MetaGPT involves tool-calling (e.g., for file writing), using GPT-4o-mini resulted in a large number of ill-structured calls and task failures. Hence, we adopted the more strong GPT-4.1-mini for MetaGPT. Therefore, the conclusion in our previous response is MaintainCoder > EvoMAC > MapCoder > AgentCoder for maintainable code generation.
>
> Thank you once again for your valuable time and constructive feedback. We promise to include these discussions in our revision. If you have any further questions or clarifications, please do not hesitate to let us know!
>
> Best wishes,
>
> The Authors

---

> > ### Author Response · Authors · 2025-08-06
> > **To Confirm Concerns Addressed & Thanks for Review Again**
> >
> > Dear Reviewer wgB7,
> >
> > Thank you so much for your quick reply and efforts on reviewing.
> >
> > After our further Response to W2, **we hope to confirm that your concerns have been addressed**.  If most of your concerns have been addressed, we respectfully request that you consider increasing the rating.
> >
> > If you have any further questions or clarifications, please do not hesitate to let us know. Thanks again!
> >
> > Best wishes,
> >
> > The Authors

---

> > > ### Comment · Reviewer_wgB7 · 2025-08-07
> > > **Response**
> > >
> > > Thanks for the additional clarification on MetaGPT's performance. The claim that MaintainCoder is akin to applying a proven, efficient algorithm holds through the agent configuration, which can indeed be effective in some specific scenarios. Most of my concerns have been addressed. I’ll keep the positive score.

---

### Official Review · Reviewer_9Ho3 · 2025-07-02

**Clarity:** 3
**Significance:** 2
**Originality:** 4
**Rating:** 4
**Confidence:** 3

**Summary:**

Taking into consideration of software maintainability, which is not very broadly discussed in previous benchmarks, the author proposed a **MaintainBench** which transforms coding questions into different yet relevant tasks, aided with human labelling. The authors further proposed **MaintainCoder** that also features an agentic pipeline that follows current human developer workflows. Results are that **MaintainCoder** outperforms baseline and certain previous methods in both *correctness* and *maintainability metrics*.

The maintainability metrics are:

  - Maintainability Index
  - Cyclomatic Complexity (CC)
  - Code Change Volume ($Code_{\textit{diff}}$)
  - Syntax Tree Similarity ($AST_\textit{sim}$)

**Questions:**

1. This paper has made a wide comparison between coding LLMs, ranging from open-source models to closed-source models and in model sizes. Reasoning models are however not included in the comparison. Could you please clarify why is the experiment lacking this category?
2. Number of tokens used are not reported in this paper. We'd like to know if the token cost is comparable to other code-gen methods or at least on a same magnitude.
3. May RL training and other non inference-time techniques benefit from your benchmark and method?
4. Can the model used for generating MaintainBench affect the results? If so, at what extent?
5. Why is there a gap between Pass@5 on APPS-Dyn and APPS?

I'm willing to increase my rating if you can provide more thorough analyses on (why your benchmark and method work), and explain why at least part of the weaknesses are so.

**Ethical Concerns:**

["NO or VERY MINOR ethics concerns only"]

**Final Justification:**

I appreciate the efforts made by the authors during the rebuttal process, especially in providing a detailed case study and detailed experiment results, and have fully addressed my concerns on the efficacy of the proposed method and token consumption. It should be worth noting that should this paper be accepted, the authors should make sure that the tables be modified to perform fair and equivalent comparisons between methods and methods, between models and models, and between methods and non-methods.

Specifically, my suggestion is that results from plain GPT, results from the very same GPT applied with your methods should appear in the same table side-by-side at least somewhere in your paper, and have it referenced in the Analysis section. With that properly addressed, I'm pretty confident that these results could support the maintainability of *MaintainCoder* at least on simple tasks.

However, one must take into consideration that the capabilities of base models are already strong enough to cover an abundance of coding tasks, and that agent-based approaches have emerged while this paper is being reviewed, it may appear necessary that *MaintainCoder* be compared with newer models over more sophisticated tasks like SWE-bench. These tasks are more challenging in terms of software complexity and maintainability and deserve more attention in potential future work.

Thus, I have changed my score from 3 to 4 (borderline accept) for this paper given its contribution to preliminary results in maintainability for LLM coding.

**Limitations:**

The authors have not thoroughly discussed the limitations of their work. We advise that the authors should address at least some of the following points (not exhaustive and open to discussions):

- (w.r.t. Weaknesses) Can methods in this benchmark be likewise applied to longer tasks (e.g. a repository's whole lifespan)?
- How would code maintainability improvements from LLM-generated code affect the overall software development process?

**Paper Formatting Concerns:**

No major formatting concerns. But please do check your paper for typos.

**Quality:**

3

**Strengths And Weaknesses:**

**Strengths:**

The authors identified that maintainability is not widely considered in current coding benchmarks and set out to resolve this issue. They leverage a custom agentic pipeline to transform existing coding problem sets into equivalent problems and *has included human-in-the-loop* to assure benchmark quality. The number of problems sum to to around 1000 questions in total.

**Weaknesses:**

The author has included a short case study w.r.t. to a MaintainBench problem, but we expect an end-to-end case study such that the entire prompts and outputs are shown, both in the 'benchmark generation' process (the problem does not necessarily has to be in the actual benchmark) and in the 'MaintainCoder' process. This is crucial to grasping 'how does your agentic pipelines work' in a more fast and concrete way. The current case study is too un-verbose and provide few insights into the details of your method.

While it may seem reasonable that, human practices, when followed by LLMs can lead to better performance in either correctness or maintainability, there is no assurance that prebuilt pipelines are the best way to achieve (large software) coding. Currently, reasoning models have large likelihoods in surpassing non-reasoning language models in multiple aspects. It is for this reason we strongly suggest the authors to include reasoning models like *o1* in the comparison, and we'd like to see that your method would still bring about some improvement atop the base models.

Since this benchmark is aimed at improving or at least measuring maintainability, we suggest that problem sets like APPS, MBPP may not suffice in terms of problem complexity. The authors should consider including more complex tasks like SWE-Bench. The method coverage is also not very broad, as only *AgentCoder* and *MapCoder* are included. We suggest that whenever time and bandwidth permits, more methods should be also included in the comparison.

---

> ### Author Rebuttal · Authors · 2025-07-31
>
> Dear Reviewer 9Ho3:
>
> Thank you for your thorough review and insightful feedback. We appreciate the opportunity to clarify our contributions and address your concerns. Below are our responses to your comments.
>
> ## Response to W1:
>
> > **(End-to-End Case Study)** The author has included a short case study w.r.t. to a MaintainBench problem, but we expect an end-to-end case study such that the entire prompts and outputs are shown, both in the 'benchmark generation' process and in the 'MaintainCoder' process.
>
> Thank you for this excellent suggestion. We are happy to provide a case study of MaintainCoder.
>
> MaintainCoder is designed to function not just as a code generator but as a software architect, creating a forward-looking architecture. **For demonstration of a 2048 game. MaintainCoder adopts a Model-View-Controller (MVC) architecture**, producing the following well-organized file structure:
>
> ```Plain
> 2048-python-game/
> ├── README.md
> ├── main.py
> └── src/
>     └── game_2048/
>         ├── __init__.py
>         │
>         ├── common/
>         │   ├── __init__.py
>         │   └── enums.py
>         │
>         ├── controller/
>         │   ├── __init__.py
>         │   └── game_controller.py
>         │
>         ├── model/
>         │   ├── __init__.py
>         │   └── game_state.py
>         │
>         ├── view/
>         │   ├── __init__.py
>         │   ├── game_view.py
>         │   └── ui_constants.py
>         │
>         └── utils/
>             ├── __init__.py
>             └── matrix_utils.py
> ```
>
> If the first requirement is "change the color of the tiles" (a View modification), and the second is "add a new '512' tile" (a Model modification), the changes are localized and simple. **This architectural foresight provides persistent benefits over multiple maintenance cycles.**
>
> ## Response to W2:
>
> > **(Reasoning Models)** We strongly suggest the authors to include reasoning models like o1 in the comparison, and we'd like to see that your method would still bring about some improvement atop the base models.
>
> Our initial experiments did not include reasoning models primarily due to high computational cost.
>
> As suggested, we conducted experiments with o3-mini. As table shown, MaintainCoder significantly **decreases maintenance costs, and improves the maintenance success rate** (Pass@5). This demonstrates the robustness and general applicability of our approach. Thank you for this suggestion.
>
> | Model                       | MI ↑ | CC ↓ | Pass@5 (%) ↑ | AST_sim ↑ | Code_diff (%) ↓ | Code_diff (abs) ↓ |
> | --------------------------- | ---- | ---- | ------------ | --------- | --------------- | ----------------- |
> | GPT-4o-mini                 | 56.5 | 6.52 | 22.5         | 0.640     | 95.3            | 21.2              |
> | MaintainCoder (GPT-4o-mini) | 64.1 | 3.11 | 30.5         | 0.812     | 28.4            | 20.5              |
> | o3-mini                     | 52.1 | 11.3 | 30.3         | 0.661     | 101.6           | 32.9              |
> | MaintainCoder (o3-mini)     | 62.3 | 3.85 | 36.4         | 0.794     | 27.8            | 21.3              |
>
> ## Response to W3:
>
> > **(More benchmarks and methods)** The authors should consider including more complex tasks like SWE-Bench. The method coverage is also not very broad, as only AgentCoder and MapCoder are included. We suggest that whenever time and bandwidth permits, more methods should be also included in the comparison.
>
> **Regarding Benchmarks:** We agree that evaluating on repository-level tasks like SWE-Bench is an important future direction. As demonstrated in Response to W1, **MaintainCoder is capable of generating repository-level code**. However, building a *maintainability* benchmark at that scale presents significant challenges in constructing and verifying the dynamic requirements and test cases. The field of code maintainability evaluation is still in its early stages. We believe **MaintainBench serves as a crucial first step, analogous to how simpler benchmarks like HumanEval and MBPP catalyzed research in code correctness**.
>
> **Regarding Methods:** As you and Reviewer wgB7 suggested, we have included more multi-agent systems. The results are shown below.
>
> *Tab2: More multi-agent baselines (MetaGPT needs gpt-4.1-mini for strong tool call capability, ChatDev has different output format and is not listed).*
>
> | CodeContest-Dyn             | MI↑  | CC↓  | pass@5↑ | AST_sim↑ | Code_diff^per↓ | Code_diff^abs↓ |
> | --------------------------- | ---- | ---- | ------- | -------- | -------------- | -------------- |
> | MaintainCoder (gpt-4o-mini) | 65.8 | 2.68 | 32.6    | 0.833    | 23.2           | 17.4           |
> | MetaGPT (gpt-4.1-mini)      | 55.2 | 7.63 | 30.3    | 0.760    | 44.3           | 18.6           |
> | EvoMAC (gpt-4o-mini)        | 62.6 | 5.18 | 26.5    | 0.685    | 60.1           | 20.0           |
>
> Thank you for pushing us to broaden the scope of our evaluation. We will add these results and a detailed discussion of related works [3, 4, 5, 6].
>
> ## Response to Q1:
>
> > **(Reasoning Models)** This paper has made a wide comparison between coding LLMs, ranging from open-source models to closed-source models and in model sizes. Reasoning models are however not included in the comparison. Could you please clarify why is the experiment lacking this category?
>
> Thank you for this question. We have provided a detailed answer in our **Response** **to W2**.
>
> ## Response to Q2:
>
> > **(Token Cost)** If the token cost is comparable to other code-gen methods or at least on a same magnitude.
>
> **Yes, the token cost is comparable to other methods like MapCoder**. We have performed the cost analysis and find that multi-agent pipelines incur higher computational costs. But for initial code generation, we believe this is a worthwhile trade-off.
>
> *Tab4: Computational overhead analysis (token usage).*
>
> | **Token Usage** | **MaintainCoder** | **MapCoder** | **o3-mini** | **MetaGPT/ChatDev** | **GPT-4o-mini** |
> | --------------- | ----------------- | ------------ | ----------- | ------------------- | --------------- |
> | CodeContests    | 33.1k             | 38.7k        | 20.8k       | 50k+                | 2.5k            |
> | xCodeEval       | 29.6k             | 23.5k        | 21.2k       | 50k+                | 2.3k            |
>
> Specifically, the token usage of MaintainCoder is on par with MapCoder or reasoning model o3-mini, and much lower than MetaGPT/ChatDev [5,6]. But **MaintainCoder achieves a >60% improvement in maintainability and higher correctness**. The initial investment is justified by the drastic reduction in subsequent human effort required for debugging and refactoring. The table below details the token consumption per stage in our pipeline.
>
> *Tab5: More fine-grained token usage.*
>
> | CodeContests | Requirement Analysis | Design Pattern Selection | Framework Design | Supervisor | Code Implementation | Code Modification | Code Extraction | Total |
> | ------------ | -------------------- | ------------------------ | ---------------- | ---------- | ------------------- | ----------------- | --------------- | ----- |
> | tokens(k)    | 1.7                  | 2.3                      | 6.7              | 6.7        | 3.1                 | 9.0               | 3.6             | 33.1  |
>
> ## Response to Q3:
>
> > **(Potential Application)** May RL training and other non inference-time techniques benefit from your benchmark and method?
>
> Thank you for this insightful question. **Yes, our work can indeed benefit RL training**. As we also discuss in our response to Reviewer AvR7, the metrics in MaintainBench (both the first-order estimates and the dynamic indicators) are designed to achieve an effective balance between **evaluation accuracy and computational efficiency**. This makes them highly suitable to be used as a reward signal for RL training, aiming at optimizing code generation models for maintainability.
>
> ## Response to Q4:
>
> > **(Model Sensitivity)** Can the model used for generating MaintainBench affect the results? If so, at what extent?
>
> We acknowledge that using GPT-4o for generating problem variations could potentially introduce a model bias. We tried our best to mitigate this, in line with best practices for synthetic data generation [1, 2]:
>
> 1. **Human Expert Validation:** Each problem variation and its corresponding solution were reviewed and refined by human experts to ensure quality and correctness.
> 2. **Execution-Based Verification:** All solutions and their evolutions are verified through execution against a comprehensive test suite.
> 3. **Traceable Co-evolution:** Solutions and test cases co-evolve, maintaining logical consistency.
>
> Most importantly, the core conclusions of our paper are drawn from **vertical comparisons** (e.g., comparing MaintainCoder (DeepSeek-V3) to the base DeepSeek-V3), not from horizontal comparisons between different model families. **These vertical experiments are robust and independent of potential benchmark bias. Therefore, this does not affect the validity of experimental conclusions.**
>
> ## Response to Q5:
>
> > Why is there a gap between Pass@5 on APPS-Dyn and APPS?
>
> Thank you for the sharp observation. The pass@5 on APPS-Dyn measures the correctness of the code *after* it has been modified to meet a new requirement based on old solution. **The evaluation target is different** from the initial code generation measured by the original APPS benchmark. Thus, it is normal to have different scores.
>
> Thanks again for the detailed comments. Your feedback is valuable for improving our paper. Let us know if there are still any concerns.
>
> [1] On LLMs-Driven Synthetic Data Generation, Curation, and Evaluation: A Survey.
>
> [2] Best Practices and Lessons Learned on Synthetic Data.
>
> [3] Metagpt: Meta programming for multi-agent collaborative framework
>
> [4] Self-evolving multi-agent collaboration networks for software development
>
> [5] Chatdev: Communicative agents for software development
>
> [6] SWE-Dev: Evaluating and Training Autonomous Feature-Driven Software Development

---

> ### Author Response · Authors · 2025-08-05
> **To Confirm Concerns Addressed & Thanks for Review**
>
> Dear Reviewer 9Ho3,
>
> As the discussion period will conclude soon, **we hope to confirm that your concerns have been addressed**.
>
> For your convenience, we have summarized our key responses below:
>
> - **Response to W1:** We provided a case study demonstrating how MaintainCoder proactively designs a maintainable MVC architecture for a 2048 game, which simplifies future updates.
> - **Response to W2:** As suggested, we have conducted new experiments with the reasoning model o3-mini, demonstrating that MaintainCoder significantly improves maintainability and correctness metrics.
> - **Response to W3:** We acknowledge the value of complex benchmarks like SWE-Bench for future work and have added more multi-agent baselines as requested, which demonstrate the superiority of MaintainCoder.
> - **Response to Q1:** We clarified that reasoning models were initially omitted due to computational cost but have now been included in our new experiments, as detailed in our response to W2.
> - **Response to Q2:** We acknowledge a higher initial token cost but argue it is a worthwhile trade-off, as it is comparable to other multi-agent systems and leads to a drastic reduction in subsequent maintenance effort.
> - **Response to Q3:** Yes, the metrics from our MaintainBench are helpful to serve as a reward signal in RL training for improved code maintainability.
> - **Response to Q4:** We mitigated potential benchmark bias to GPT series through human validation and execution-based verification, and our main conclusions are unaffected as they rely on vertical comparisons between a base model and its MaintainCoder-enhanced version.
> - **Response to Q5:** The performance gap exists because APPS-Dyn evaluates the code's correctness after maintenance, which is a fundamentally different from the initial code generation task in the original APPS benchmark.
>
> We have spared no effort in addressing your concerns, and remain open to discussing any further questions or clarifications. Thank you again for improving our paper!
>
> Best wishes,
>
> The Authors

---

> > ### Author Response · Authors · 2025-08-08
> > **To Confirm Concerns Addressed & Thanks for Review Again**
> >
> > Dear Reviewer 9Ho3,
> >
> > As the discussion period concludes in one day, **we hope to confirm that your concerns have been addressed** (or at least largely).
> >
> > To further address your W3, we supplement the comparison between multi-agent systems below. As shown, multi-agent systems focused on problem-solving do not necessarily improve code maintainability over the base model (e.g., AgentCoder). **In stark contrast, MaintainCoder explicitly incorporates the wisdom of long-term human engineering practices—classic design patterns—and achieves significant optimization in maintainability**.
> >
> > *Exp. The comparison between multi-agent systems*
> >
> > | CodeContest-Dyn             | MI↑  | CC↓  | pass@5↑ | AST_sim↑ | Code_diff^per↓ |
> > | --------------------------- | ---- | ---- | ------- | -------- | -------------- |
> > | GPT-4o-mini                 | 57.8 | 6.06 | 24.2    | 0.661    | 90.1           |
> > | MaintainCoder (GPT-4o-mini) | 65.8 | 2.68 | 32.6    | 0.833    | 23.2           |
> > | EvoMAC (GPT-4o-mini)        | 62.6 | 5.18 | 26.5    | 0.685    | 60.1           |
> > | AgentCoder (GPT-4o-mini)    | 62.5 | 7.28 | 18.2    | 0.629    | 44.9           |
> >
> >
> >
> > If our clarifications have resolved the main points you raised, we would be grateful if you would consider reassessing your rating. Thank you for your valuable time and consideration.
> >
> > Best regards,
> >
> > The Authors

---

> > > ### Comment · Reviewer_9Ho3 · 2025-08-08
> > > **Confirmation of Concerns Addressed**
> > >
> > > I appreciate the efforts made by the everyone during the rebuttal process. I have already changed my score from 3 to 4 (borderline accept) for this paper accordingly given its contribution to preliminary results in maintainability for LLM coding.
> > >
> > > My suggestion is that results from plain GPT, results from the very same GPT applied with your methods should appear in the same table side-by-side at least somewhere in your paper, and have it referenced in the Analysis section. With that properly addressed, I'm pretty confident that these results could support the maintainability of *MaintainCoder* at least on simple tasks.
> > >
> > > The authors should consider evaluating *MaintainCoder* on new, agentic LLMs over SWE-bench and other more complex tasks, so as to provide more evidence in cases closer to real-world software development scenarios.

---

> > > > ### Author Response · Authors · 2025-08-09
> > > > **Response to Suggestions & Thanks for Review**
> > > >
> > > > Dear Reviewer 9Ho3,
> > > >
> > > > We appreciate your constructive suggestions and the encouraging rating.
> > > >
> > > > In Tab. 1 and 2, we have already included results from plain GPT, GPT_CoT, GPT_Plan and multi-agent systems on the very same GPT in the same tables. As suggested, we will supplement more multi-agent systems (MetaGPT, EvoMAC, SWE-agent, ...) beyond existing AgentCoder and MapCoder.
> > > >
> > > > We will continue focusing on maintainable code generation, and advancing the scope to agentic LLMs over more real-world software development scenarios like SWE-bench.
> > > >
> > > > Finally, we thank you again for the valuable suggestions!
> > > >
> > > > Best regards,
> > > >
> > > > The Authors

---

### Official Review · Reviewer_kobS · 2025-07-02

**Clarity:** 4
**Significance:** 4
**Originality:** 4
**Rating:** 4
**Confidence:** 4

**Summary:**

This paper introduces a benchmark and evaluation criteria to assess code generation taking into maintainability account. Their proposed approach includes an orchestrated pipeline of LLM agents. The agents are specialized in four software development phases and takes into account contextual awareness in inter-agent communication via AutoGen.

**Questions:**

Please refer to the weakness Section. I a happy to raise my score if my questions are addressed.

**Ethical Concerns:**

["NO or VERY MINOR ethics concerns only"]

**Final Justification:**

I have read all the authors' rebuttal. I would like to keep my score unchanged.

**Limitations:**

Please see my comments on weakness.

**Quality:**

3

**Strengths And Weaknesses:**

Strength

1.	Proposes a maintainability metrics to measure

2.	Extensive evaluation on correctness and maintainability

3.	Performance boost over high performing models in optimizing second round of code generation with modified requirements (Section 4.1). Previous models optimized for number of lines of codes which may lead to unmaintainable codes in the long run. The proposed method generates code with minimized inflated lines.

Weakness

1.	In Section 3.2.1, “We sample 30 problems from each dataset randomly, and extend them to 120 new problems by systematically modifying their requirements”
Could the authors elaborate on their generation scheme of requirements?

2.	Section 3.2.2: how the authors evaluate solution-test co-evaluation?

3. The code generation and code optimization agents need detailed description.n. Please see details below.

3a. The code generation agent.
“After the initial generation, a test sample selected from the test set is inserted as insertion for iterative debugging”.
Generating high quality test cases is a challenging software engineering problem. Did the authors generate new test cases for each four-requirement generation in Section 3.2.2?

3b. The code optimization agent contributes to most performance boost, as demonstrated in ablation test in Table 4. It would be helpful to have detailed analysis of the agent.
In the code optimization agent in Section 3.2.2, the authors mentioned “…iterative process continues until the code meets requirements and pass tests, ensuring functional, efficient, and reliable solutions”
 Could the authors elaborate on
1)	Number of test cases
2)	Number of iterations
3)	How they measured efficiency

---

> ### Author Rebuttal · Authors · 2025-07-31
>
> Dear Reviewer kobS:
>
> Thank you for your careful review and insightful feedback. We greatly appreciate the time and effort you have invested. Let us address your questions one by one.
>
> ## Response to W1:
>
> > **(Requirement Generation Details)** In Section 3.2.1, "We sample 30 problems from each dataset randomly, and extend them to 120 new problems by systematically modifying their requirements" Elaborate more on generation scheme of requirements.
>
> To ensure both diversity and quality of MaintainBench, we implemented a structured, multi-stage process for requirement changes:
>
> - For each initial problem, we systematically applied **four distinct requirement change patterns** using carefully designed prompts to simulate diverse real-world software maintenance scenarios.
> - To guarantee the quality of the resulting dataset, we followed a rigorous filtering protocol. We initially sampled over 50 problems from the source datasets. We then filtered out any problem that failed to produce a valid and meaningful new problem for all four change patterns.
> - Subsequently, these generated problems underwent a **manual evaluation** by human annotators to ensure their correctness, clarity, and relevance. This stringent process resulted in the final set of 30 high-quality base problems, which were extended to the 120 new problems used in our benchmark.
>
> We have provided a detailed description of the change patterns and the generation pipeline in Appendix A/B/C. As suggested, we will add a more detailed scheme summary to Section 3.2.1 in the revision. Thanks!
>
> ## Response to W2:
>
> > **(Solution-Test Co-evolution Details)** In Section 3.2.2, how to evaluate solution-test co-evolution?
>
> Thanks for your question. We clarify that solution-test co-evolution is fully integrated and evaluated.
>
> - Firstly, we adapt the original solution code according to a specific requirement change pattern and perform a first-round check to ensure that the adapted code satisfies our formatting and structural conventions.
> - Secondly, based on the same requirement change, we simultaneously generate a new problem `new_problem`, a new solution `new_solution`, and a corresponding test case `new_test_case`.
> - Thirdly, human annotators manually inspect the `new_problem` to ensure that it is appropriate and that it faithfully reflects the intended requirement change; any problem that fails to meet this criterion is discarded.
> - Fourthly, for the remaining valid cases, annotators not oly execute `new_solution` on `new_test_case`, but also manually verify the correctness of both the `new_solution` and `new_test_case`, making necessary corrections when issues are identified.
>
> This multi-stage pipeline ensures coherence between the problem, solution, and test case, with both automated checks and human validation in this solution-test co-evolution process. Please refer to Appendix B.2 for more details. As suggested, we will make more clarifications in the revision.
>
> ## Response to W3:
>
> > **(Code Generation and** **Optimization** **Details)** For iterative debugging, did the authors generate new test cases for each four-requirement generation in Section 3.2.2? For code optimization, please elaborate on 1) Number of test cases 2) Number of iterations 3) How to measure efficiency.
>
> Thank you for these important questions about our experimental setup. We will address them sequentially.
>
> **Test Cases for Iterative Debugging:** Yes. For each new problem, we generated a corresponding new set of test cases. This process involved both model-based generation and meticulous human verification. The goal was to ensure that the new test suite comprehensively covers both the newly introduced functionalities and the original functionalities.
>
> **Details of Code** **Optimization****:** For the iterative code optimization and debugging phase, our setup is as follows:
>
> - **Number of Test Cases:** The model uses **one** test case, randomly sampled from the newly generated test suite for that specific problem. This test case provides the feedback (i.e., pass or fail signal) that guides the model's subsequent refinement attempts. This setup is consistent with the evaluation protocols in prior works on code generation, such as AgentCoder and MapCoder  [1,2].
> - **Number of Iterations & Efficiency:** We set the maximum number of optimization iterations to **five**. This decision was based on empirical analysis and the goal of balancing performance improvement against computational cost (efficiency). As table shown, performance generally improves with more iterations. We observe a significant boost within the first three iterations, with performance gains starting to saturate around the fifth iteration. Limiting the process to five steps prevents excessive resource consumption while capturing the majority of the potential performance gains, a trade-off that aligns with previous research [1,2].
>
> *Tab 1: Performance vs. Number of* *Optimization* *Iterations (**opt**).*
>
> | Dataset      | Model                      | opt=0 | opt=1 | opt=2 | opt=3 | opt=4 | opt=5 |
> | ------------ | -------------------------- | ----- | ----- | ----- | ----- | ----- | ----- |
> | CodeContests | MaintainCoder(gpt-4o-mini) | 28.0  | 28.6  | 31.7  | 30.4  | 32.6  | 32.6  |
> |              | MaintainCoder(deepseek-v3) | 25.7  | 28.6  | 27.6  | 30.4  | 32.6  | 37.9  |
> | xCodeEval    | MaintainCoder(gpt-4o-mini) | 17.2  | 18.0  | 19.2  | 23.4  | 25.0  | 27.3  |
> |              | MaintainCoder(deepseek-v3) | 25.0  | 24.1  | 27.9  | 30.8  | 32.6  | 36.7  |
>
> We hope these responses have adequately addressed your concerns. Let us know if there are still any concerns!
>
> ## Reference
>
> [1] MapCoder: Multi-Agent Code Generation for Competitive Problem Solving
>
> [2] AgentCoder: Multi-Agent-based Code Generation with Iterative Testing and Optimisation

---

> > ### Comment · Reviewer_kobS · 2025-08-07
> >
> > I would like to thank the authors for their responses. However, there are still some concerns: for W1 and W2, the authors performed sampling but I'm not sure whether those samples are representative of the entire dataset.
> >
> > Additionally, I would recommend authors to consult software engineering papers regarding generating test cases. Some of the criteria are: including boundary values, including un-acceptable values, all pairs test, and others. Random sampling does not guarantee this.
> >
> > I will keep my score unchanged.

---

> ### Author Response · Authors · 2025-08-05
> **To Confirm Concerns Addressed & Thanks for Review**
>
> Dear Reviewer kobS,
>
> As the discussion period will conclude soon, **we hope to confirm that your concerns have been addressed**.
>
> For your convenience, we have summarized our key responses below:
>
> - **Response to W1 (Requirement Generation):**
>
>   We generated new requirements by systematically applying four distinct change patterns to each base problem, followed by a rigorous filtering process and manual validation to ensure high quality and relevance.
>
> - **Response to W2 (Solution-Test Co-evolution):**
>
>   Solution-test co-evolution is ensured by concurrently generating the new solution and test case, which are then manually inspected and verified by human annotators for mutual correctness and alignment with the new requirements.
>
> - **Response to W3 (Code** **Optimization****):**
>
>   We generated a new, human-verified test suite for each new problem. For iterative optimization, the model uses feedback from a single randomly sampled test case across a maximum of five iterations, a limit empirically chosen to balance performance gains with computational efficiency.
>
> We have spared no effort in addressing your concerns, and remain open to discussing any further questions or clarifications. Thank you again for improving our paper!
>
> Best wishes,
>
> The Authors

---

> ### Author Response · Authors · 2025-08-07
> **Further Response & Thanks for Review**
>
> Dear Reviewer kobS,
>
> Thank you first for the careful review and feedback. We are glad to hear that some of your concerns have been addressed. Please allow us to provide further clarifications.
>
> ------
>
> ## Representativeness of Sampled Datasets
>
> For better representativeness of sampled datasets, **we have paid special attention to the coverage of task difficulty during the construction of MaintainBench**. MaintainBench aggregates five sub-datasets that cover a range of problems from easy to hard (the dynamic versions of HumanEval, MBPP, APPS, CodeContests, xCodeEval). E.g., we provide the distribution of difficulty levels on APPS-Dyn, demonstrating our selected samples cover multiple difficulty tiers. Note that, the APPS dataset is of mixture-level difficulty.
>
> |             | Introductory Level | Interview Level | Competition Level |
> | :---------- | :----------------- | :-------------- | :---------------- |
> | **APPS-Dyn** | 30%                | 28%             | 42%               |
>
> Furthermore, without random sampling, applying four types of requirement changes to all original problems would result in new datasets four times the original size, leading to enormous annotation costs. As you mentioned, generating high quality test cases is a challenging software engineering problem. Objectively, manually annotating a large volume of data can easily lead to a decrease in annotation quality. **Therefore, we opted to annotate a smaller, yet sufficient, number of samples for benchmark, to maximize the annotation quality for these randomly sampled instances**. We acknowledge that due to human and financial budgets, the scale of the dataset could be further expanded to improve its representativeness of the original dataset. However, it is worth noting that even the *HumanEval dataset proposed by OpenAI involves only 164 Python problems* with limited test cases, and subsequent works like EvalPlus have expanded the scale of test cases and problems [1,2]. **MaintainBench is the first benchmark for maintainable code generation, and we believe our initial effort is yet of significant importance**.
>
> ------
>
> ## Test Case Synthesis
>
> We acknowledge that high-quality test case synthesis is challenging [3,4]. Despite randomly generated by LLMs, our test case synthesis involves quality control in three aspects:
>
> * First and foremost, we have included the test cases from the original dataset (appended to the raw solution as assert statements) in the synthesis prompts. Therefore, **the model can modify these original test cases to build new ones for the new problem, thereby inheriting, to some extent, the quality level of the original test cases (e.g., coverage).**
> * Second, during the **manual review phase**, we place special emphasis on the effectiveness of the test cases, **requiring that every requirement change includes corresponding test cases** to ensure the modification has a detectable effect.
> * Third, for the prompt of error handling enhancements in A.1, we explicitly require the model to generate tests for the locations of the requirement changes, such as, "The test input should trigger errors that need to be addressed in the new problem as much as possible."
>
> Although our current work does not hard-code software engineering testing adequacy criteria (such as all-pairs, boundary value analysis, etc.), these three aspects provide a certain degree of quality assurance for the test cases. We are grateful for your feedback and suggestion, and promise to elaborate these details in our revision.
>
> ------
>
> We appreciate your valuable review and constructive feedback. If you have any further questions or clarifications, please do not hesitate to let us know!
>
> Best wishes,
>
> The Authors
>
> ## Reference
>
> [1] Is Your Code Generated by ChatGPT Really Correct? Rigorous Evaluation of Large Language Models for Code Generation
>
> [2] CodeContests+: High-Quality Test Case Generation for Competitive Programming
>
> [3] Software testing with large language models: Survey, landscape, and vision
>
> [4] A systematic review of the application and empirical investigation of search-based test case generation

---

> > ### Author Response · Authors · 2025-08-08
> > **To Confirm Concerns Addressed & Thanks for Review Again**
> >
> > Dear Reviewer kobS,
> >
> > We appreciate your feedback and the opportunity to provide further clarification.
> >
> > We have posted further response regarding the *representativeness of sampled datasets* and *test case synthesis*, and **we hope it has addressed your concerns** (E.g., our new test cases inherit the quality and coverage of the original test cases to some extent).  If our clarifications have resolved the main points you raised, we would be grateful if you would consider reassessing your rating.
> >
> > Thank you for your valuable time and consideration.
> >
> > Best regards,
> >
> > The Authors

---

### Official Review · Reviewer_AvR7 · 2025-07-04

**Clarity:** 3
**Significance:** 3
**Originality:** 3
**Rating:** 5
**Confidence:** 4

**Summary:**

This paper introduces MaintainCoder, a framework for generating maintainable code under dynamic requirements, and MaintainBench, the first benchmark for evaluating code maintainability. Key contributions include:

1. MaintainBench: Extends established benchmarks (e.g., HumanEval, MBPP) with 500+ Python tasks, covering four requirement change patterns (*functional extensions, interface modifications, data structure transformations, error handling enhancements*). Novel dynamic metrics (e.g., post-modification correctness `Pass@k`, code change volume `Code_diff`, AST similarity `AST_sim`) are proposed.
2. MaintainCoder: A multi-agent system integrating the Waterfall model and classical design patterns. It enforces modularity, low coupling, and high cohesion through specialized agents (e.g., requirements analysis, pattern selection, framework evaluation).
3. Key Findings:
   - Existing methods (GPT-4o, AgentCoder, etc.) suffer significant maintainability degradation under requirement changes (e.g., `Pass@5` drops 15–30%).
   - MaintainCoder improves dynamic metrics by >60% while boosting initial correctness.
   - Traditional static metrics (e.g., MI, cyclomatic complexity) poorly correlate with maintainability, while dynamic metrics show high consistency.

**Questions:**

N/A

**Ethical Concerns:**

["NO or VERY MINOR ethics concerns only"]

**Final Justification:**

Regarding the responses to W1–W4, with the exception of the Human Baseline issue, the authors have provided fairly comprehensive experimental evidence addressing my concerns. Considering my earlier score of 4 (“Borderline Accept”) and their response, I am now inclined to support this paper and get the final acceptance decision.

**Limitations:**

Yes

**Quality:**

2

**Strengths And Weaknesses:**

Strengths
1. Problem Significance:
   - Addresses the critical yet underexplored challenge of maintainability in code generation, aligning with real-world pain points (80% of software costs stem from maintenance).
   - Highlights limitations of static benchmarks (e.g., HumanEval) and advocates for dynamic evaluation.
2. Novel Contributions:
   - MaintainBench:
     - Systematically models requirement evolution via four change types (`Π_B`), covering 80% of maintenance scenarios (adaptive/perfective).
     - Ensures quality via solution-test co-evolution (GPT-4o + human refinement).
   - MaintainCoder:
     - Uniquely combines software engineering principles (Waterfall model, design patterns) with multi-agent collaboration.
     - Ablation studies validate the necessity of framework evaluation and code optimization modules (25–37% performance drop if removed).
3. Rigorous Evaluation:
   - Comprehensive Coverage: Tests 3 difficulty levels (entry/mixture/competition), 7 base models (GPT-4o, DeepSeek-V3, etc.), and 2 multi-agent frameworks (AgentCoder, MapCoder).
   - Multi-dimensional Metrics: Exposes discrepancies between static metrics (MI/CC) and dynamic maintainability (e.g., high MI but low `Pass@5`).
   - Robustness Checks: Consistent results across varying `Pass@k` (k=1–5) and Phase II generators.

Weaknesses

1. Theoretical Gaps in Dynamic Metrics:
   - The Monte Carlo estimation for maintenance cost (`ℳ(C₀)`) lacks justification for first-order truncation. Discount factor `γ` is not discussed.
   - `AST_sim` and `Code_diff` rely on syntactic similarity (via `difflib`), ignoring semantic equivalence (e.g., refactored but functionally identical code).
2. Limited Experimental Comparisons:
   - Missing Human Baseline: No comparison against code written/adapted by developers, leaving industrial relevance unclear.
   - Language Restriction: Only Python is evaluated; generalization to stricter-typed languages (e.g., Java) is untested.
3. Methodological Limitations:
   - Black-box Pattern Selection: Design pattern choices rely on LLM reasoning without causal analysis (only cites correlation in [13]).
   - Computational Cost: Multi-agent pipeline requires multiple LLM calls (≤3 framework evaluations + ≤5 optimizations), but overhead vs. single-model inference is unreported.
4. Dataset Issues:
   - Annotation Process: Appendix B.3 mentions "multi-stage expert review" but omits reviewer count, disagreement resolution, or inter-rater agreement (e.g., Kappa).
   - Class Balance: Distribution of change types (`Π_B`) across difficulty levels is unanalyzed.

---

> ### Author Rebuttal · Authors · 2025-07-30
>
> Dear Reviewer AvR7:
>
> Thank you so much for your constructive feedback and efforts on our research.
>
> We supplement several experiments to address your concerns.
>
> ## Response to W1:
>
> > **(Theoretical Gaps in Dynamic Metrics)**
> >
> > - The Monte Carlo estimation for maintenance cost lacks justification for first-order truncation.
> > - AST_sim and Code_diff rely on syntactic similarity (via difflib), ignoring semantic equivalence (e.g., refactored but functionally identical code).
>
> We appreciate your rigorous feedback on our dynamic metrics.
>
> **First-Order Truncation for** **Maintenance** **Cost:**
>
> We agree that real-world maintenance is a long-term, multi-round process. However, simulating multi-round maintenance for evaluation is computationally prohibitive. We believe that **code maintainability is an intrinsic property**, and thus the single-round maintenance cost should be highly correlated with the cumulative cost of multiple rounds. **A single round, therefore, can serve as an effective and efficient proxy.**
>
> To validate this, we followed your suggestion and extended our benchmark. We synthesized a second round of requirement changes for MaintainBench. As shown below, the costs from the **second round** **exhibit a high correlation with the first-round costs**, and are slightly more than the first ones. This suggests that the first-order approximation provides a good balance between evaluation accuracy and computational feasibility. We will supplement this discussion in the revision.
>
> *Tab1: First-round and Second-round* *Maintenance* *Costs (AST_sim & Code_diff) on CodeContests-Dyn.*
>
> | Model                       | 1st Round Cost |                 | 2nd Round Cost |                 |
> | --------------------------- | -------------- | --------------- | -------------- | --------------- |
> |                             | AST_sim ↑      | Code_diff (%) ↓ | AST_sim ↑      | Code_diff (%) ↓ |
> | GPT-4o-mini                 | 0.661          | 90.1            | 0.645          | 96.2            |
> | MaintainCoder (GPT-4o-mini) | 0.833          | 23.2            | 0.811          | 29.4            |
> | DeepSeek-V3                 | 0.718          | 87.2            | 0.694          | 92.5            |
> | MaintainCoder (DeepSeek-V3) | 0.788          | 43.2            | 0.765          | 48.9            |
>
> **Syntactic vs. Semantic Metrics:**
>
> We acknowledge that `AST_sim` and `Code_diff` do not capture the semantic equivalence of refactored code. However, we argue that this is a feature, not a bug. **The effort required for refactoring functionally identical code is a genuine part of the maintenance cost that our metrics are designed to capture**. A developer still needs to spend time and cognitive energy to perform such refactoring. While more semantically-aware metrics are a valuable direction for future research, syntax-based metrics like `difflib` offer the unique advantages of being fast and easily verifiable, which is more friendly for methods involving reinforcement learning (e.g., RLVR).
>
> ## Response to W2:
>
> > **(Limited Experimental Comparisons)**
> >
> > - Human Baseline: No comparison against code written/adapted by developers.
> > - Language Restriction: Only Python is evaluated; stricter-typed languages (e.g., Java) is untested.
>
> Thanks for important suggestions to broaden our evaluation scope.
>
> **Human Baseline:** Based on the suggestion, we have already begun the process of recruiting experienced developers to perform coding. Given the time required for recruitment, coding, and data analysis, we commit to providing these results during the discussion period as soon as possible. Thanks!
>
> **Language Generalization (Java):** For industry, design patterns are successful and widely adopted in Java software development. For experiments, we translated MaintainBench from Python to Java, and evaluated MaintainCoder against the baseline on Java version. The results are consistent with Python: **MaintainCoder improves maintainability metrics, indicating our principles are not language-specific**. We also find Java performance is generally slightly lower than Python, due to imbalance training data [1,2]. We will supplement these discussion in the revision.
>
> *Tab2: Java version performance on CodeContests-Dyn.*
>
> | Model                       | MI ↑ | CC ↓ | Pass@5 (%) ↑ | AST_sim ↑ | Code_diff (%) ↓ | Code_diff (abs) ↓ |
> | --------------------------- | ---- | ---- | ------------ | --------- | --------------- | ----------------- |
> | GPT-4o-mini                 | 56.5 | 6.52 | 22.5         | 0.640     | 95.3            | 21.2              |
> | MaintainCoder (GPT-4o-mini) | 64.1 | 3.11 | 30.5         | 0.812     | 28.4            | 20.5              |
> | DeepSeek-V3                 | 54.2 | 7.21 | 25.0         | 0.703     | 91.8            | 23.8              |
> | MaintainCoder (DeepSeek-V3) | 62.5 | 3.84 | 36.0         | 0.795     | 45.1            | 22.7              |
>
> ## Response to W3:
>
> > **(Methodological Limitations)**
> >
> > - Black-box Pattern Selection: Design pattern choices rely on LLM reasoning without causal analysis.
> > - Computational Cost: Multi-agent pipeline requires multiple LLM calls (≤3 framework evaluations + ≤5 optimizations). The overhead vs. single-model inference is needed.
>
> **"Black-box" Pattern Selection:** We clarity that our pattern selection is not purely "black-box." Unlike monolithic, end-to-end models, our multi-agent pipeline makes this reasoning an **explicit and auditable** step. The Pattern Selection Agent is required to first propose a design pattern and then provide a textual justification for its choice based on the requirements. This output is reflected and logged, offering a degree of transparency and interpretability. While if LLM reasoning meets causal analysis is a big open-question beyond the scope of this paper, our method makes the *process* more transparent, mimicking auditable teamwork.
>
> **Computational Cost:** We have performed the requested analysis and agree that multi-agent pipelines incur higher computational costs. But for initial code generation, we believe this is a **worthwhile trade-off**. Specifically, the token usage of MaintainCoder is on par with MapCoder or reasoning model o3-mini, and much lower than MetaGPT/ChatDev [3,4]. But **MaintainCoder achieves a >60% improvement in maintainability and higher correctness**. The initial investment is justified by the drastic reduction in subsequent human effort required for debugging and refactoring.
>
> *Tab3: Token Usage Comparison.*
>
> | **Token Usage** | **MaintainCoder** | **MapCoder** | **o3-mini** | **MetaGPT/ChatDev** | **GPT-4o-mini** |
> | --------------- | ----------------- | ------------ | ----------- | ------------------- | --------------- |
> | CodeContests    | 33.1k             | 38.7k        | 20.8k       | 50k+                | 2.5k            |
> | xCodeEval       | 29.6k             | 23.5k        | 21.2k       | 50k+                | 2.3k            |
>
> ## Response to W4:
>
> > **(Dataset Issues)**
> >
> > - Annotation Process: Appendix B.3 mentions "multi-stage expert review" but omits reviewer count, disagreement resolution, or inter-rater agreement.
> > - Class Balance: Distribution of change types across difficulty levels is unanalyzed.
>
> **Annotation Process Details:** Appendix B.3 mentions a “multi-stage expert review,” which refers to our sequential quality control pipeline. In this process, each problem was assigned to a qualified annotator (32 Solution-Test Co-evolution per annotator) responsible for correcting test inputs and ensuring compliance with the four evolution patterns. Subsequently, two of the authors independently reviewed the annotations. Disagreements, when present, were resolved through discussion to reach consensus. After annotation and review, the authors conducted comprehensive validation to ensure that all test cases execute correctly within our evaluation framework, the requirement evolution patterns are properly implemented, and the new problems represent meaningful software maintenance scenarios. To further assess annotation reliability, we computed inter-annotator agreement using Fleiss’ Kappa between the two reviewing authors from our team, obtaining a score of 0.899 [5]. This value indicates almost perfect agreement, suggesting that our annotation process is consistent and reliable across different contexts. We will include these clarifications in our final revision.
>
> **Class Balance Analysis:** In MaintainBench, every original problem has four requirement changes. The distribution of change types is fully balanced, ensuring that our evaluation is not biased towards a specific type of change or difficulty. For some fine-grained analysis, we provide the distribution of error changes across difficulty levels.
>
> *Tab3: The distribution of error changes across difficulty levels.*
>
> | Difficulty  | Functional Extension | Interface Mod. | Data Struct. Trans. | Error Handling |
> | ----------- | -------------------- | -------------- | ------------------- | -------------- |
> | Entry       | 28%                  | 23%            | 26%                 | 23%            |
> | Mixture     | 26%                  | 24%            | 25%                 | 25%            |
> | Competition | 24%                  | 27%            | 22%                 | 27%            |
>
> Finally, thanks again for the valuable suggestions, which have helped us improve the quality of our work.  Hope our responses address your concerns. Please feel free to contact us if you have more questions or suggestions!
>
> ## Reference
>
> [1] ClassEval: A Manually-Crafted Benchmark for Evaluating LLMs on Class-level Code Generation
>
> [2] JavaBench: A Benchmark of Object-Oriented Code Generation for Evaluating Large Language Models
>
> [3] MapCoder: Multi-Agent Code Generation for Competitive Problem Solving
>
> [4] AgentCoder: Multi-Agent-based Code Generation with Iterative Testing and Optimisation
>
> [5] Fleiss' Kappa: Measuring Agreement Among Multiple Raters

---

> ### Author Response · Authors · 2025-08-05
> **Response to Human Baseline**
>
> Dear Reviewer AvR7,
>
> Thank you again for your constructive feedback. Following your suggestion, we have now incorporated a human performance baseline.
>
> ## Response to Human Baseline:
>
> > **(Human Baseline)** No comparison against code written/adapted by developers.
>
> For CodeContests-Dyn, we have recruited participants with experience in ICPC-style competitions, holding Codeforces ratings between 1700 and 2300. The participants are required to finish coding in 30 minutes per problem.
>
> As shown below, the code from human programmers, demonstrates poorer maintainability metrics than AI-generated codes. **It highlights that even skilled developers can produce less maintainable code, particularly under problem-solving pressure (very common in real-life scenarios)**, thereby underscoring the significance and practical necessity of MaintainCoder to systematically improve this crucial software quality attribute - maintainability.
>
> | CodeContests-Dyn            | MI ↑     | CC ↓     | Pass@5 (%) ↑ | AST_sim ↑ | Code_diff (%) ↓ |
> | --------------------------- | -------- | -------- | ------------ | --------- | --------------- |
> | **Human**                   | **53.2** | **8.17** | **23.5**     | **0.541** | **112.3**       |
> | GPT-4o-mini                 | 57.8     | 6.06     | 24.2         | 0.661     | 90.1            |
> | MaintainCoder (GPT-4o-mini) | 65.8     | 2.68     | 32.6         | 0.833     | 23.2            |
> | DeepSeek-V3                 | 55.7     | 6.80     | 26.5         | 0.718     | 87.2            |
> | MaintainCoder (DeepSeek-V3) | 63.1     | 3.64     | 37.9         | 0.788     | 43.2            |
>
> We hope this additional experiment and our response have adequately addressed your concerns. We appreciate the feedback, which has helped us strengthen our paper.
>
> Best wishes,
>
> The Authors

---

> ### Author Response · Authors · 2025-08-05
> **To Confirm Concerns Addressed & Thanks for Review**
>
> Dear Reviewer AvR7,
>
> As the discussion period will conclude soon, **we hope to confirm that your concerns have been addressed**.
>
> For your convenience, we have summarized our key responses below:
>
> - **Response to W1:** We validate that the single-round maintenance cost is an effective proxy for long-term cost through new multi-round experiments showing high correlation. We also clarify that our maintenance metrics intentionally capture refactoring efforts as part of the maintenance cost, offering accuracy and efficiency.
> - **Response to W2:** We conducted new experiments on Java, demonstrating our method's principles generalize beyond Python. And we supplemented human baseline, highlighting that even skilled developers can produce less maintainable code, particularly under problem-solving pressure.
> - **Response to W3:** We clarify that our pattern selection process is transparent and auditable, not a black-box, and justify our computational cost is on par with other multi-agent systems and worthwhile for substantially improved maintainability.
> - **Response to W4:** We have detailed our rigorous annotation process, which achieved a high inter-rater agreement (Fleiss' Kappa = 0.899), and confirmed that the requirement change types in our benchmark are intentionally balanced.
>
> We have spared no effort in addressing your concerns, and remain open to discussing any further questions or clarifications. Thank you again for improving our paper!
>
> Best wishes,
>
> The Authors

---

### Author Response · Authors · 2025-08-04
**Global Summary and Response to Area Chairs and All Reviewers**

Dear Area Chairs and Reviewers,

We are very grateful for the reviewers' recognition of our work, such as:

**Reviewer AvR7:**
> Addresses the critical yet underexplored challenge of maintainability in code generation, aligning with real-world pain points (80% of software costs stem from maintenance).

**Reviewer kobS:**
> Extensive evaluation on correctness and maintainability. Previous models optimized for number of lines of codes which may lead to unmaintainable codes in the long run. The proposed method generates code with minimized inflated lines.

**Reviewer 9Ho3:**
> The authors identified that maintainability is not widely considered in current coding benchmarks and set out to resolve this issue.

**Reviewer wgB7:**
> The work provides both immediate practical value and opens promising research directions, potentially influencing how the community evaluates code generation systems.

Furthermore, **both reviewers kobS and 9Ho3 kindly expressed their willingness to increase the ratings if the concerns are addressed**. We have carefully addressed all the reviewers' concerns point by point, with extra experiments correspondingly provided. The core contributions, responses to common issues and paper supplements are as follows:

## Core Contributions

- **Dynamic Benchmark.** We introduce MaintainBench, the first benchmark assessing code maintainability through requirement evolution cycles. Constructed through systematic extension of well-established benchmarks (HumanEval, MBPP, APPS, CodeContest, xCodeEval), it incorporates diverse requirement changes with expert-curated test cases. Novel metrics quantify modification efforts dynamically.
- **Maintainable Generation.** We introduce MaintainCoder as a pioneering solution. It integrates the waterfall model with multi-agent collaboration and classical design patterns to enforce high cohesion, low coupling, single responsibility, and maintainability.
- **Empirical Insights.** Extensive experiments reveal that (1) Previous methods suffer from significant maintainability degradation under dynamic requirements (low pass rate and high code modification), even for multi-agent systems. (2) MaintainCoder not only improves maintainability metrics by up to +60\%, but also enhances the correctness of initial codes. (3) Existing static metrics not only fail to accurately reflect maintenance efforts, but also exhibit contradictory trends among different metrics. In contrast, dynamic metrics from MaintainBench are more effective and consistent.

## Responses to Common Issues
- **Generalizability of Languages and Base Models:** Beyond Python benchmarks and non-reasoning models, reviewers suggested to test our framework across different languages and against more advanced methods. In response, we have conducted new experiments by:
    1.  Translating part of MaintainBench into **Java** version, confirming that MaintainCoder's benefits are language-agnostic.
    2.  Benchmarking against a powerful **reasoning model (o3-mini)** and more **multi-agent systems (MetaGPT, EvoMAC)**. The experiments show that MaintainCoder not only yields significant maintainability enhancements, but also improves on top of reasoning base models.

- **Long-Term Maintenance Evaluation:** As maintainability is an intrinsic property of codes, we adopted single-round evaluation as a proxy for maintenance costs. The reviewers asked for more justification. In response, while simulating multi-round evolution is computationally prohibitive, we synthesized a second round of requirement changes. **The experiments showed a high correlation between first and second-round maintenance costs, supporting that the single-round cost is an effective and efficient proxy.**

- **Token Usage:** We provided a detailed breakdown of token usage, showing that **MaintainCoder's cost is comparable to other multi-agent systems like MapCoder and lower than frameworks like MetaGPT/ChatDev**. We argue that this initial computational investment is a worthwhile trade-off, justified by the significant reduction in downstream human maintenance efforts.


## Paper Supplements

Most reviews focused on clarifying details or discussing further improvements. We will supplement our responses in the paper:
- More Justification on Long-Term Maintenance Proxy [Method, Appendix (Reviewer AvR7, wgB7)]
- Experiments on Language Generalization (Java) [Appendix (Reviewer AvR7, wgB7)]
- Expanded Comparisons with Reasoning Models and More Multi-Agent Systems [Experiments, Related Work (Reviewer 9Ho3, wgB7)]
- Detailed Analysis of Token Usage [Experiments (Reviewer AvR7, 9Ho3, wgB7)]
- More Details on the MaintainBench (E.g. Inter-Annotator Agreement, Solution-Test Co-evolution) and MaintainCoder (E.g. Iterative Debugging, Case Study) [Appendix]

We sincerely thank all the reviewers for their valuable and constructive comments, which greatly help us improve the quality of our paper.

Looking forward to further discussions with the reviewers!

---

> ### Author Response · Authors · 2025-08-09
> **Thank You for Your Engagement**
>
> Dear Area Chairs and Reviewers,
>
> We are glad to hear that most of concerns from reviewers have been addressed during discussion.
>
> As the response period draws to a close, we wanted to thank you all again for the time and effort in improving our paper!
>
> Best regards,
>
> Authors

---

### Note · Authors · 2025-08-12

Dear Area Chairs and Reviewers,

We appreciate your active engagement and invaluable reviews. After discussion, Reviewer wgB7, kobS and AvR7 keep their positive scores. Reviewer 9Ho3 has increased the rating from 3 to 4, leading to a consensus of positive evaluation.

We are glad to hear that most of concerns from reviewers have been addressed. We promise to supplement our clarifications and expanded experiments (token usage, more multi-agent systems, etc) in the revision, and plan more complex tasks like SWE-Bench as our future work.

Thank you once again for your time and efforts on our work!

Best regards,

The Authors

---

### Decision · Program_Chairs · 2025-09-17

**Decision:**

Accept (poster)

**Comment:**

- Summary: This paper focuses on maintainability in code generation and proposes MaintainCoder as a multi-agent system to generate maintainable code under dynamic requirements integrating Waterfall model and classical design patterns. The paper also introduces MaintainBench, extension of common benchmarks for code generation by subjecting the code generation task to different requirements. They also propose novel metrics capturing various aspects of maintenance efforts.
- Strengths:
  - The paper explores a significant problem in code generation, motivated by real world scenarios and would be of interest to the Neurips community.
  - The experiments conducted are rigorous, spanning various base models, difficulty levels, agent frameworks.
  - The paper’s contributions on the benchmark and the agent system are novel.
- Weaknesses:
  - The reviewers mentioned the lack of evaluations with respect to human written code. Even though the authors tried to address this during the rebuttal, I feel like the design of this would benefit from a round of peer review. The reviewers mentioned other limitations of evaluations like generalization to different languages, reasoning models and the authors addressed these concerns during the rebuttal phase.
  - Application of the approach to SWE-bench like benchmarks would be interesting.
- Suggestions:
  - It is suggested by reviewer 9Ho3 and wgB7 to make the comparisons with raw models and models paired with agentic approaches more clear as part of the results.
- Recommendation:
  - All of the four reviewers recommended acceptance of the paper, three of whom gave borderline acceptance scores.
  - This paper tackles an underexplored area in code generation and provides benchmarks to motivate further research in the area. I believe it will be of interest to the Neurips community and there is why I recommend acceptance.